# RESOURCE EFFICIENT TEST-TIME TRAINING WITH SLIMMABLE NETWORK

## ABSTRACT

Test-Time Training (TTT), an innovative paradigm for enhancing a model's generalization in a specific future scenario, commonly leverages self-supervised learning to adapt the model to the unlabeled test data under distribution shifts. However, previous TTT methods tend to disregard resource constraints during the deployment phase in real-world scenarios and have two fundamental shortcomings. Firstly, they are obligated to retrain adapted models when deploying across multiple devices with diverse resource limitations, causing considerable resource inefficiency. Secondly, they are incapable of coping with computational budget variations during the testing stage. To tackle these issues, we propose a resource-adaptive test-time training framework called SlimTTT, which allows for the seamless switching of different sub-networks for adaptive inference. Furthermore, we discover that different width of sub-networks can capture different views of images and these views are complementary and beneficial to the ones created by data augmentation, which is widely used in TTT. To utilize these views, we introduce Width-enhance Contrastive Learning (WCL), Logits Consistency Regularization (LCR) and Global Feature Alignment (GFA) to promote representation consistency at both feature and prediction space in a self-supervised manner, enabling networks of different widths to excel in TTT tasks. Our proposed method, SlimTTT, has achieved state-of-the-art (SOTA) results across a variety of adaptation methods and four different datasets with varying backbones. Remarkably, despite a significant reduction in computational complexity - over $70\%$ less than the current SOTA method - SlimTTT continues to deliver competitive performance, rendering it highly conducive for adoption in practice.

## 1 INTRODUCTION

Generalizing deep learning models to new domains holds significant importance, and numerous existing methods strive to train models to exhibit robustness against potential distribution shifts encountered during training (Ganin & Lempitsky, 2015; Long et al., 2018; Ilse et al., 2020). Recently, a novel adaptation protocol known as Test-Time Training (TTT) (Sun et al., 2020; Liu et al., 2021; Su et al., 2022; Gandelsman et al., 2022) seeks to improve a model's generalization performance in a specific future scenario where it is deployed or tested under distribution shifts. TTT achieves this goal by optimizing a self-supervised objective using unlabeled test data at test time, making it more practical for realistic applications. Nevertheless, having an extra training stage after the model deployment makes the new protocol highly dependent on computational resources (e.g., GPU memory, peak memory) that can vary widely across different devices, yet none of the existing TTT methods have taken the resource constrains into consideration. Consequently, it is imperative to explore Resource Efficient Test-Time Training algorithms that can tackle a larger range of practical scenarios.

Test-Time Training (TTT) typically involves three phases: training-time training, test-time training, and inference. Here exists two realistic scenarios with computational constraints as depicted in Figure 1(left). First, **the computational resources differ across devices**, and thus the training-time trained model must meet the requirements of each device to enable on-device test-time training after deployment (Chakraborty et al., 2023; Meng et al., 2022; Yang et al., 2020). Second, **after deployment, the available computational budget of a same device may change during inference** (Yu et al., 2018; Yu & Huang, 2019b;a) (for example, the power saving mode will reduce available computing capacity). For the first scenario, a vanilla solution would be to pretrain an individual model

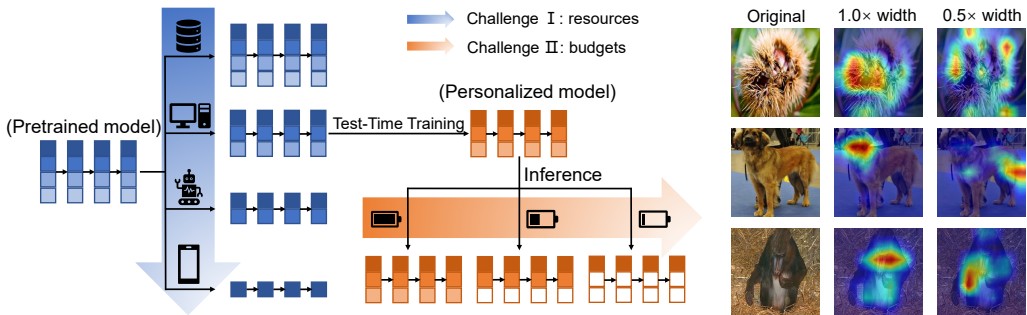

Figure 1: **Left**: Slimmable network addresses two practical resource-aware scenarios of test-time training. After training-time training, different width of pretrained slimmable network (blue) can be deployed according to devices' resource constraints. During Inference, the personalized slimmable network (orange) can be switched to different width according to the real-time limitation of the device. **Right**: The visualization of the attention of sub-networks of different widths using Grad-CAM. Different sub-networks pay attention to different parts of the object, which visually confirms that they capture multiple views of the image.

for each device that is tailored to its specific limitations. However, apart from being expensive, such method fails to manage the second scenario under TTT protocol: when device's resources become limited during inference, a smaller pretrained model must be redeployed and then updated all over again using test data. Therefore, to conduct more applicable Test-Time Training in both scenarios, we seek for networks that can operate with their sub-networks instantly and adaptively without requiring retraining, and design TTT algorithms specifically tailored to these networks.

In this paper, motivated by the characteristic of anytime neural networks (Li et al., 2022; Cai et al., 2019; Yu et al., 2020; Yu & Huang, 2019b), we proposes to leverage the typical slimmable network (Yu et al., 2018) to achieve resource efficient Test-Time Training. Conceptually, slimmable network offers a solution to the challenges outlined above as depicted in Figure 1. After training a single supernet, its sub-networks with different widths can be deployed depending on the device's resource limitations. Furthermore, each sub-network is considered a device-specific supernet and is further updated using test data after deployment, which allows all sub-networks to collectively enhance generalization performance. During inference, these sub-networks can be flexibly switched to achieve optimal accuracy-efficiency trade-off, depending on the device's status.

Unfortunately, directly applying existing TTT methods to a slimmable architecture yields no improved or even worse results due to the change in the network structure. To promote resource efficient Test-Time Training, we intensively study one characteristic of the slimmable network, which is its capability of capturing multiple views of the data simultaneously, and then explicitly enhance semantic consistency learning using such property. Specifically, this architecture enables the execution of sub-networks with different width, and these sub-networks produce distinct representations of the same input data, laying the foundation for their knowledge interaction during test-time training process. Here, we provide a visual demonstration and a toy example to support for key observation.

Visually, the Grad-CAM visualization (Selvaraju et al., 2017) in Figure 1(right) shows the attention of sub-networks of different widths, where we can observe that the regions of interest for networks with widths of $1.0\times$ and $0.5\times$ are different, which demonstrates that networks of different widths learn different views of the same inputs. Mathematically, we build connection to the previous work (Allen-Zhu & Li, 2023) which proves that differently initialized networks can capture different views of data. We use a three-layer MLP as a toy example to show that sub-networks in the slimmable architecture can be regard as the full network with different initializations. Our full MLP model contains two input neurons, three hidden layer neurons and one output neuron. Its sub-network drops a neuron in the hidden state. Given input data $\boldsymbol{x}$, the sub-network output $y_{sub}$ and the full output $y_{sup}$ can be represented as:

$$y_{sub} = \tilde{\mathbf{A}}\left(\sigma(\tilde{\mathbf{B}}\boldsymbol{x})\right) = \begin{bmatrix} a_{11} & a_{12} \end{bmatrix} \left(\text{ReLU}(\begin{bmatrix} b_{11} & b_{12} \\ b_{21} & b_{22} \end{bmatrix}\begin{bmatrix} x_1 \\ x_2 \end{bmatrix})\right), \quad (1)$$

$$y_{sup} = \begin{bmatrix} a_{11} & a_{12} & a_{13} \end{bmatrix} \left(\text{ReLU}(\begin{bmatrix} b_{11} & b_{12} \\ b_{21} & b_{22} \\ b_{31} & b_{32} \end{bmatrix}\begin{bmatrix} x_1 \\ x_2 \end{bmatrix})\right) = \begin{bmatrix} \tilde{\mathbf{A}} & a_{13} \end{bmatrix} \left(\text{ReLU}(\begin{bmatrix} \tilde{\mathbf{B}} \\ \boldsymbol{b}_3 \end{bmatrix}\begin{bmatrix} x_1 \\ x_2 \end{bmatrix})\right) \quad (2)$$

It is evident that when $a_{13} = 0$ and $\boldsymbol{b}_3 = \boldsymbol{0}$, we have $y_{sub} = y_{sup}$, indicating the equivalance in output between the sub-network and the full network with certain parameter changes.

Expanding upon our findings, we develop a new SlimTTT framework based on comprehensive multi-view consistency learning, which includes a Width-enhanced Contrastive loss and Logit Consistency Regularization. These strategies fully exploit information within multiple data views captured by the sub-networks at both feature and logit space. Additionally, an Resource-Aware Ensemble strategy is proposed, aimimg to improve inference performance without increasing the required inference resources beyond acceptable time limits. We demonstrate the superiority of our method, through comparing with various TTT methods on standard benchmarks including ImageNet-C, CIFAR10-C, CIFAR10.1, and CIFAR100-C with different backbones such as ResNet50, MobileNetV1, MobileNetV2 and ViT. In a nutshell, our contributions can be summarized as:

- We investigate a more challenging yet realistic Test-Time Training scenario by taking resource constrains into consideration, and propose a general solution using the representative architecture of slimmable neural network. Moreover, we exploit the architecture of slimmable network and find that different width of sub-networks can capture multiple views of the same input data, possessing a talent for enhancing TTT performance.

- We propose a resource efficient SlimTTT framework that includes Width-enhanced Contrastive Learning, Logit Consistency Regularization and Global Feature Alignment in test-time training phase and Resource-Aware Ensemble in inference phase. Our framework not only addresses the pratical challenges of TTT, but also encourages multi-view consistency that enhance the model's generalization performance.

- Comprehensive experimental results demonstrate that SlimTTT consistently achieves superior performance on ImageNet-C and other standard benchmarks compared to other competitive methods that use similarly scaled backbones. Notably, even with a computing complexity reduced by $70\%$, our method still can outperform most of other methods.

## 2 RELATED WORK

**Anytime Neural Network**  Anytime neural networks (Huang et al., 2018; Yu et al., 2018; Li et al., 2022; Yang et al., 2020; Cai et al., 2019; Yu et al., 2020) are elaborately designed architectures that contain various sub-networks within a single network. These sub-networks can be trained jointly and executed independently, allowing the single supernet can be deployed instantly and adaptively under different resource constraints to address practical scenarios in real-world applications. According to the structure of the sub-networks, anytime neural networks can be generally divided into networks with variable depth (Huang et al., 2018; Cai et al., 2019; Yu et al., 2020; Larsson et al., 2016) and variable width (Lee & Shin, 2018; Yu et al., 2018; Li et al., 2021a). Slimmable network (Yu et al., 2018; Yu & Huang, 2019b) is a typical architecture with variable width. The original slimmable network (Yu et al., 2018) uses switchable batch normalization and in-place distillation to guarantee the sub-networks' performance in any width. Building upon prior research, DSNet (Li et al., 2021a) dynamically adjusts filter numbers of networks at test time with respect to different inputs. MutualNet (Yang et al., 2020) uses varying input resolutions to train slimmable networks, allowing each sub-network to learn multi-scale representations in a mutually beneficial manner. However, they all assume that training and test data follow the same distribution, and the model's performance may significantly drop once the distribution differs.

**Test-Time Training/Adaptation**  Improving the generalization capabilities of models to effectively address domain shift has been a prominent focus in deep learning. Many research efforts have emerged under settings such as Domain Adaptation (DA) (Tzeng et al., 2014; Long et al., 2015; Ganin & Lempitsky, 2015; Long et al., 2018; Li et al., 2020), Source-Free Domain Adaptation (Liang et al., 2020b;a), Domain Generalization (Yue et al., 2019; Balaji et al., 2018; Ilse et al., 2020), etc. Recently, Sun et al. first propose Test-Time Training (TTT) (Sun et al., 2020) as a novel approach towards domain shift by updating the model parameters using test samples in an unsupervised manner before actually making a prediction. This method incorporates a self-supervised branch (rotation prediction) and only updates the feature encoder using self-supervised loss in test-time training process. The subsequent works TTT++ (Liu et al., 2021) and TTAC (Su et al., 2022) improve the test-time training performance by replacing the rotation prediction objective with contrastive learning objective (Chen et al., 2020) to learn more informative representations.

Test-Time Adaptation (TTA) is a similar setting to TTT that further considers the case where source supervision is not available. With only the source pretrained model, Test-time Normalization (Ioffe & Szegedy, 2015) updates the affine parameters in batch normalization using the test samples, Tent (Wang et al., 2020) adopts entropy minimization across a batch of test data and T3A (Iwasawa & Matsuo, 2021) maintains a number of support sets. The latter approaches MEMO (Zhang et al., 2022) and TPT (Shu et al., 2022) both utilize a various kinds of strong data augmentation and minimize the entropy of the ensemble of model prediction across all the augmented images.

However, these TTT/TTA methods address the domain-shift challenge in an ideal scenario, while in realistic applications, various constraints, particularly computational resource limitations, need to be taken into consideration. This paper considers resource limitation of edge devices and proposes SlimTTT using slimmable network to achieve the optimal accuracy-efficiency trade-off on different devices in Test-Time Training.

**Resource Efficient Domain Adaptation**    This work also takes inspiration from resource-efficient domain adaptation, which focuses on saving computational resources of domain adaptation models (Jiang et al., 2020; Li et al., 2021b; Meng et al., 2022; Chakraborty et al., 2023). DDA (Li et al., 2021b) and REDA (Jiang et al., 2020) propose efficient DA algorithms based on MSDNet (Huang et al., 2018), whereas SlimDA (Meng et al., 2022) and AnyDA (Chakraborty et al., 2023) improve the adaptation performance using slimmable networks (Yu et al., 2018; Yang et al., 2020). Compared to these approaches addressing domain adaptation problems, our resource efficient Test-Time Training is more flexible, which no longer anticipates a certain target domain in advance but adapts the model depending on the test samples. Moreover, our SlimTTT takes fully advantage of the Slimmable Network structure by jointly exploiting different views of the data produced by both sub-networks and augmentation using a comprehensive contrastive learning framework.

## 3    METHOD

In this section, we will provide a comprehensive explanation of our methodology to enable resource-efficient Test-Time Training. To begin with, we formalize the problem for both the training-time training phase and test-time training phase. Then, a joint training approach following standard practice (Sun et al., 2020; Liu et al., 2021) is introduced for the training-time training phase. Moving on to the test-time training phase, we propose three techniques to facilitate multi-view consistency learning for better adaptation, namely Width-enhanced Contrastive Learning (WCL), Logits Consistency Regularization (LCR) and Global Feature Alignment (GFA). Finally, we elaborate on an ensemble approach for inference. The key section of test-time training is illustrated in Figure 2.

### 3.1    PROBLEM DEFINITION AND NOTATIONS

Test-Time Training incorporates three stages: training-time training, test-time training and inference. In the first stage, a model is trained on the labeled source dataset $D_s = \{\boldsymbol{x}^s, y^s\}$ and then deployed. In the second stage, test-time training, the deployed model adapts to the test distribution according to the unlabeled test set $D_t = \{\boldsymbol{x}^t\}$, which could be sampled from a different domain. Since the source dataset and the test set are not simultaneously available, for ease of notations, we omit the subscript $s$, $t$ on the data and denote them as $\boldsymbol{x}$ uniformly in the rest of this section. The goal of Resource Efficient Test-Time Training is to maximize the model's inference performance without exceeding the resource constraint $\mathcal{B}$.

Each slimmable network adopted in this paper can be considered as a set of $K$ sub-networks of different width. We denote the slimmable feature encoder $g$, main classification head $\pi_m$ and self-supervised head $\pi_s$ as $l = \{l^{(1)}, l^{(2)}, ..., l^{(K)}\}_{l \in \{g, \pi_m, \pi_s\}}$, where network with larger index has larger width ($l^{(K)}$ denotes the supernet). For the $k$-th sub-network, we denote $\boldsymbol{p}_i^{(k)} = softmax(\pi_m^{(k)} \circ g^{(k)}(\boldsymbol{x}_i))$ its prediction of sample $\boldsymbol{x}_i$ with $softmax(\cdot)$ being the standard softmax function, and $\boldsymbol{h}_i = \pi_s^{(k)} \circ g^{(k)}(\boldsymbol{x}_i)$ its projected feature.

### 3.2    TRAINING-TIME TRAINING PHASE

To facilitate the model training during test-time with unlabeled data, we follow standard practice (Sun et al., 2020; Liu et al., 2021) here in the training-time and consider a joint training process

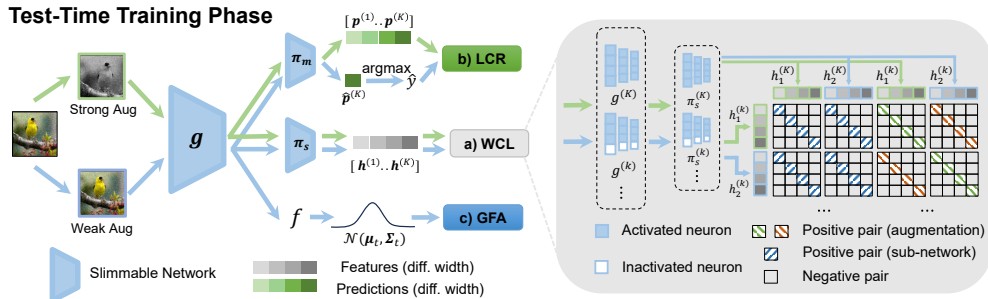

Figure 2: Overview of the SlimTTT framework which includes three techniques: **a) WCL** the projections of two augmented views of each width of network are combined with the biggest network's projections to encourage the consistency between both multiple augmented views and multiple width views. **b) LCR** The weak augmented version is sent to the biggest network to obtain a pseudo label, which supervises the predictions of each width of the sub-networks. **c) GFA** The global test data distributions of different width are aligned to the source one of the same width using KL-Divergence.

that involves a main training objective and an auxiliary self-supervised learning task. For a batch of $N$ labeled images in $D_s$, the main training loss function $\mathcal{L}_{main}$ can be expressed as:

$$\mathcal{L}_{main} = \sum_{i=1}^{N} (\sum_{k=1}^{K} \mathcal{E}(\boldsymbol{p}_i^{(k)}, y_i) + \sum_{k=1}^{K-1} \mathcal{E}(\boldsymbol{p}_i^{(k)}, \boldsymbol{p}_i^{(K)})), \tag{3}$$

where $\mathcal{E}(\cdot, \cdot)$ is the cross-entropy loss. The first term minimizes the empirical risk and the second term improves the prediction consistency between sub-networks and the supernet and is only minimized with respect to each sub-network.

The self-supervised loss $\mathcal{L}_s$ is a width-enhanced contrastive loss which will be introduced in detail in § 3.3. Overall, the joint training loss $\mathcal{L}_{joint}$ is:

$$\mathcal{L}_{joint} = \mathcal{L}_{main} + \mathcal{L}_s. \tag{4}$$

### 3.3 TEST-TIME TRAINING PHASE

In test-time training phase, we fully exploit the slimmable network's capability of capturing multiple data views to improve its performance. Specifically, our framework consists of three parts: WCL and LCR that encourage the multi-view consistency on both feature and logit level, and GFA that apply a restriction over the multi-view features. Next, we provide detailed explanations and discussions of these techniques.

#### 3.3.1 WIDTH-ENHANCED CONTRASTIVE LEARNING

As discovered in early experiments (Figure 1(b)), sub-networks of different width in the slimmable network can capture different aspects of the same test input, which enables multi-view consistency learning without requiring the label. Therefore, upon the common practice of contrastive learning that only encourages feature consistency between differently augmented views, we propose WCL (Figure 2a) that comprehensively exploits data-views generated from both data augmentation techniques and slimmable network structures. In this regard, our model benefit from stronger supervision brought by increased variety of data views (Tian et al., 2020).

For a randomly sampled batch of $N$ images from the test dataset $D_t$, we begin by obtaining two views for each sample by applying two random data augmentations, resulting in $2N$ data views as the batch input for the slimmable network, which finally yields a total of $2KN$ representations. Since the network with full width is most powerful, we consider the $2N$ views created by the largest network as the "core views" (Tian et al., 2020), and the features created by other width of network are all encouraged to make consistency with the "core views". In this case, each sub-networks can have knowledge interaction with the largest network.

More specifically, for each of the $K-1$ sub-networks except the biggest one, we combine its $2N$ representations $\{\boldsymbol{h}_i^{(k)}\}_{i=1}^{2N}$ with those embedded by the biggest network $\{\boldsymbol{h}_i^{(K)}\}_{i=1}^{2N}$, resulting in $4N$

features $H_{all} = \{\boldsymbol{h}_i^{(k)}, \boldsymbol{h}_i^{(K)}\}_{i=1}^{2N}$. We define the positive set and negative set for each representation $\boldsymbol{h}_i^{(k)}$ as $H_{pos} = \{\boldsymbol{h}_i^{(k)}, \boldsymbol{h}_j^{(k)}, \boldsymbol{h}_i^{(K)}, \boldsymbol{h}_j^{(K)}\}$ and $H_{neg} = H_{all} \setminus H_{pos}$, where $\boldsymbol{h}_j^{(k)}$ is generated from another augmentation. Given that now we have multiple positive pairs for each representation, we adopt the multi-positive NCE loss (Lee et al., 2022), which can avoid interference between positive pairs:

$$\mathcal{L}_{sd,i} = -\sum_{k=1}^{K-1} \sum_{\boldsymbol{h}_p \in H_{pos}} \log \frac{\exp\left(\delta(\boldsymbol{h}_i^{(k)}, \boldsymbol{h}_p)/\tau\right)}{\exp\left(\delta(\boldsymbol{h}_i^{(k)}, \boldsymbol{h}_p)/\tau\right) + \sum\limits_{\boldsymbol{h}_n \in H_{neg}} \exp\left(\delta(\boldsymbol{h}_i^{(k)}, \boldsymbol{h}_n)/\tau\right)}, i = 1, ..., 2N, \tag{5}$$

where $\delta(\cdot, \cdot)$ denotes the cosine similarity. Additionally, we ensure the consistency between two augmented views created by the largest network $\langle \boldsymbol{h}_i^{(K)}, \boldsymbol{h}_j^{(K)} \rangle$ with the following contrastive loss:

$$\mathcal{L}_{s,i} = -\log \frac{\exp\left(\delta(\boldsymbol{h}_i^{(K)}, \boldsymbol{h}_j^{(K)})/\tau\right)}{\exp\left(\delta(\boldsymbol{h}_i^{(K)}, \boldsymbol{h}_j^{(K)})/\tau\right) + \sum\limits_{m=1}^{2N} \mathbb{1}_{[m \neq i, m \neq j]} \exp\left(\delta(\boldsymbol{h}_i^{(K)}, \boldsymbol{h}_m^{(K)})/\tau\right)}, i = 1, ..., 2N. \tag{6}$$

Finally, the WCL loss for the whole batch is expressed as:

$$\mathcal{L}_s = \sum_{i=1}^{2N} (\mathcal{L}_{s,i} + \mathcal{L}_{sd,i}). \tag{7}$$

***Discussion***: We find empirically that the automatically obtained views from varing width of sub-networks contain complementary information to data views manually generated by data augmentation techniques, where the latter part is widely adopted in recent TTT and TTA methods to boost adaptation performance (Liu et al., 2021; Su et al., 2022; Zhang et al., 2022). Hence, we argue that the slimmable network should be highly favorable for conducting TTT. See appendix for supports.

### 3.3.2 LOGITS CONSISTENCY REGULARIZATION

The Width-enhanced Contrastive Loss can be regarded as encouraging multi-view consistency within feature embedding space created by the auxiliary head $\pi_s$. To further encourage the multi-view consistency in prediction space, we additionally present Logits Consistency Regularization strategy (Figure 2b)). Particularly, a weakly augmented view of the test sample $\boldsymbol{x}_\alpha$ is sent to the largest network to obtain a prediction $\hat{\boldsymbol{p}}^{(K)} = softmax(\pi_m^{(K)} \circ g^{(K)}(\boldsymbol{x}_\alpha))$. Then the pseudo-label defined as $\hat{y} = \arg\max_c(\hat{p}_c^{(K)})$ is used to regularize the prediction consistency among all sub-networks. The LCR loss is defined as:

$$\mathcal{L}_d = \sum_{i=1}^{N} \left( \sum_{k=1}^{K} \mathcal{E}(p_i^{(k)}, \hat{y}_i) \right). \tag{8}$$

During optimization, the main classification head $\pi_m$ is kept frozen to prevent overfitting. In this way, the slimmable encoders are actually trained to capture invariant features across multiple views.

***Discussion***: The significance of the additional supervision from the main branch during TTT is largely ignored by previous TTT methods. TTT-R (Sun et al., 2020), TTT++ (Liu et al., 2021) and TTAC (Su et al., 2022) all construct supervise signal solely on the auxiliary branch, leading to the possibility of feature deterioration with respect to the main task. The proposed LCR loss, on the contrary, balances the supervision from both branches during the TTT optimization process.

### 3.3.3 TOTAL OBJECTIVE

Besides the two strategies for multi-view consistency learning above, the global feature alignment (GFA) loss $\mathcal{L}_a$ (Su et al., 2022) is applied to ensure that the test features do not deviate far from the source feature distribution during the whole adaptation process and it can apply a restriction over the multi-view features for each width of sub-networks and ensure the correctness of the multi-view representations. Combining all three training objectives with trade-off parameters $\lambda_s, \lambda_d$ and $\lambda_a$, the total loss in test-time training phase can be expressed as follows.

$$\mathcal{L} = \lambda_s \mathcal{L}_s + \lambda_d \mathcal{L}_d + \lambda_a \mathcal{L}_a. \tag{9}$$

### 3.4 RESOURCE-AWARE ENSEMBLE OF OUTPUTS FOR INFERENCE

This technique is based on the principle (Allen-Zhu & Li, 2023) that each model can learn different views of an image, and combining these views can result in a better prediction. In our case, different widths sub-networks within the slimmable network can capture different views of an image. To fully leverage the abundant multi-view knowledge of the slimmable network, we provide a strategy of ensembling outputs of different width within the slimmable network. For the $k^{th}$ sub-network, its final prediction from assembled outputs is: $\boldsymbol{p}_{ens}^{(k)} = \sum_{j=1}^{k} \boldsymbol{p}^{(j)}$. Note that the ensemble process will not violate the resource constraints, given that all the outputs aggregated are produced by sub-networks smaller than the current one.

## 4 EXPERIMENTS

In this section, we describe four datasets that are utilized to evaluate our method. We conduct a comparative study between our method using various sub-networks ($1.0\times, 0.75\times, 0.5\times, 0.25\times$) within the slimmable network and other TTT/TTA methods using ResNet models (Res101, Res50, Res34, Res18) of comparable sizes on different datasets. Additionally, we present ablation and analytic experiments to investigate impact of different network components and running cost analysis.

### 4.1 DATASETS AND SETUP

To assess the effectiveness of our method on **CIFAR10-C/CIFAR100-C** (Hendrycks & Dietterich, 2019), which consists of 10/100 classes with 50,000 training samples and 10,000 corrupted test samples, we pretrain the slimmable network from scratch on **CIFAR10/CIFAR100** (Krizhevsky et al., 2009). In addition, we examine the performance of our approach on a challenging dataset **CIFAR10.1** (Recht et al., 2019), which comprises approximately 2,000 difficult testing images collected over several years of research on the original **CIFAR10**. To evaluate on large-scale corrupted test samples, we fine-tune the pretrained slimmable network (Yu et al., 2018) on **ImageNet** (Deng et al., 2009) and use **ImageNet-C** (Hendrycks & Dietterich, 2019), which consists of 1,000 classes with 50,000 corrupted test samples for test-time training. The implementation details of different datasets and backbones can be found in Appendix.

### 4.2 OVERALL RESULTS

Our approach, slimTTT, utilizes Res50 (He et al., 2016) as the backbone, incorporating four switchable widths ($1.0\times, 0.75\times, 0.5\times, 0.25\times$). We conduct experiments comparing SlimTTT to various TTT and TTA methods using Res101/50/34/18 as the backbone. Specifically, we compare multiple widths of SlimTTT against comparable backbone scales of other methods, in order to ensure a fair comparison. The detailed results of different datasets can be found in Appendix.

In Figure 3, we employ a range of network widths as the largest network during the test-time training phase, which allows us to simulate different resource constraints of diverse devices. For instance, if a device has a FLOPs constraint of less than 4.0G, we can only use a $0.75\times$ (2.3G) network width as the biggest network and apply our method during test-time training. As a result, the error rate of the $0.75\times$ network width is $45.41\%$ on the ImageNet-

Figure 3: Error rate ($\%$) of our method and other TTT/TTA method base on resource constraints on the **ImageNet-C** (left) and **CIFAR100-C** (right) datasets. Lower is better.

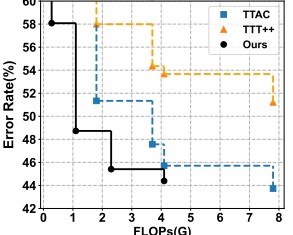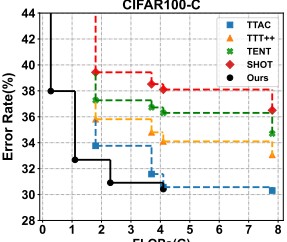

C dataset and $30.91\%$ on the CIFAR100-C dataset. In the same situation, other TTT/TTA methods can only use Res34 (3.7G) as the backbone and our method's performance is better than all of them. Remarkably, our approach achieves superior performance with fewer FLOPs on both datasets. For example, our method using the $0.5\times$ (1.1G) network width as the biggest network even can outperform most of other TTT/TTA methods using Res101 (7.8G) as the backbone.

Table 1: Error rate (%) comparison between our method taking $1.0\times$ width of network as the biggest network and different adaptation methods on **ImageNet** and **CIFAR** datasets. **ImageNet-C Avg**, **C10-C Avg**, **C10.1 Avg** and **C100-C Avg** refer to the average error rate in percentage of 15 corruption task in level 5 severity on **ImageNet-C**, **CIFAR10-C**, **CIFAR10.1** and **CIFAR100-C**. The best results are in bold black lettering.

|  | Method | Backbone | #Params | #FLOPs | ImageNet-C Avg. | C10-C Avg. | C10.1 Avg. | C100-C Avg. |
|---|---|---|---|---|---|---|---|---|
| 25.6M #Params | TEST | R-50 | 25.6M | 4.1G | 80.70 | 29.10 | 12.10 | 59.20 |
|  | BN (Ioffe & Szegedy, 2015) | R-50 | 25.6M | 4.1G | 63.04 | 15.70 | 14.10 | 43.30 |
|  | TTT-R (Sun et al., 2020) | R-50 | 25.6M | 4.1G | - | 14.30 | 11.00 | 40.40 |
|  | SHOT (Liang et al., 2020a) | R-50 | 25.6M | 4.1G | 51.59 | 14.70 | 11.10 | 38.10 |
|  | TENT (Wang et al., 2020) | R-50 | 25.6M | 4.1G | 56.89 | 12.60 | 13.40 | 36.30 |
|  | TTT++ (Liu et al., 2021) | R-50 | 25.6M | 4.1G | 53.67 | 9.80 | 9.50 | 34.10 |
|  | TTAC (Su et al., 2022) | R-50 | 25.6M | 4.1G | 45.71 | 8.52 | 9.20 | 30.57 |
|  | **SlimTTT (w/o ensemble)** | R-50 | 25.6M | 4.1G | 44.39 | 8.86 | 9.05 | 30.42 |
|  | **SlimTTT (w/ ensemble)** | R-50 | 25.6M | 7.8G | **43.98** | **8.33** | **8.95** | **29.36** |
| 14.7 - 21.8M #Params | TEST | R-34 | 21.8M | 3.7G | 80.23 | 29.10 | 10.75 | 59.00 |
|  | SHOT(Liang et al., 2020a) | R-34 | 21.8M | 3.7G | - | 10.57 | 12.75 | 38.52 |
|  | TENT(Wang et al., 2020) | R-34 | 21.8M | 3.7G | - | 11.88 | 12.55 | 36.72 |
|  | TTT++(Liu et al., 2021) | R-34 | 21.8M | 3.7G | 54.35 | 9.83 | 9.90 | 34.80 |
|  | TTAC(Su et al., 2022) | R-34 | 21.8M | 3.7G | 47.57 | 8.69 | 9.85 | 31.57 |
|  | **SlimTTT (w/o ensemble)** | R-50$_{[0.75\times]}$ | **14.7M** | **2.3G** | 45.08 | 9.37 | 9.65 | 31.11 |
|  | **SlimTTT (w/ ensemble)** | R-50$_{[0.75\times]}$ | **14.7M** | 3.7G | **44.77** | **8.63** | **9.25** | **30.34** |
| 6.9 - 11.9M #Params | TEST | R-18 | 11.9M | 1.8G | 84.53 | 31.44 | 12.85 | 60.10 |
|  | SHOT(Liang et al., 2020a) | R-18 | 11.9M | 1.8G | - | 11.27 | 13.05 | 39.43 |
|  | TENT(Wang et al., 2020) | R-18 | 11.9M | 1.8G | - | 12.53 | 13.00 | 37.27 |
|  | TTT++(Liu et al., 2021) | R-18 | 11.9M | 1.8G | 57.99 | 10.73 | 10.05 | 35.81 |
|  | TTAC(Su et al., 2022) | R-18 | 11.9M | 1.8G | 51.34 | 9.72 | 11.50 | 33.75 |
|  | ITTA(Chen et al., 2023) | R-18 | 11.9M | 1.8G | - | 10.52 | 11.20 | 34.53 |
|  | **SlimTTT (w/o ensemble)** | R-50$_{[0.5\times]}$ | **6.9M** | **1.1G** | 47.24 | 9.62 | 10.45 | 33.05 |
|  | **SlimTTT (w/ ensemble)** | R-50$_{[0.5\times]}$ | **6.9M** | 1.4G | **47.05** | **9.12** | **9.85** | **32.07** |
| 2.0M | **SlimTTT** | R-50$_{[0.25\times]}$ | **2.0M** | **278M** | **53.60** | **10.68** | **11.25** | **37.71** |

In Table 1, we take $1.0\times$ width of network as the biggest network to show the best performance each width can achieve. Table 1 shows the average error rate in percentage of 15 corruption tasks in level 5 severity on ImageNet-C, CIFAR10-C, CIFAR10.1 and CIFAR100-C. Our method achieves superior performance compared to prior SOTA TTT/TTA methods on each dataset. It is worth noting that the ensemble strategy leads to an increase in FLOPs but results in a significant performance improvement. Moreover, even without using the ensemble technique, the performance of SlimTTT can still surpass existing TTT/TTA methods. Remarkably, even with smaller parameters and FLOPs for the $0.5\times$ and $0.75\times$ width of SlimTTT, they exhibit better performance than other TTT or TTA methods that use Res34 and Res18 as backbones.

## 4.3  ABLATION STUDY

**Ablation study for individual components.** We conduct the first ablation experiment on CIFAR10-C to study the impact of individual components. The results presented in Table 2 show that Global Feature Alignment (GFA) is a fundamental strategy to apply restriction of the multiple views. Adding Width-enhanced Contrastive Learning (WCL) resulted

Table 2: The average error rate (%) results of ablation study for individual components on **CIFAR10-C** dataset.

| Component | SlimTTT | | | | | | | | |
|---|---|---|---|---|---|---|---|---|---|
| WCL | - | - | - | - | ✓ | ✓ | ✓ | - | ✓ |
| LCR | - | - | - | ✓ | - | ✓ | - | ✓ | ✓ |
| GFA | - | - | ✓ | ✓ | ✓ | ✓ | ✓ | ✓ | ✓ |
| Ensemble | - | ✓ | - | - | - | - | ✓ | ✓ | ✓ |
| R-50 | 29.53 | 27.62 | 11.61 | 9.83 | 8.99 | 8.86 | 8.68 | 8.96 | **8.33** |
| R-50$_{[0.75\times]}$ | 30.85 | 28.00 | 12.44 | 10.44 | 9.82 | 9.37 | 9.14 | 9.43 | **8.63** |
| R-50$_{[0.5\times]}$ | 29.19 | 28.60 | 13.77 | 10.84 | 10.08 | 9.62 | 10.00 | 10.22 | **9.12** |
| R-50$_{[0.25\times]}$ | 31.63 | 31.63 | 15.41 | 12.06 | 12.05 | 10.71 | 12.33 | 12.27 | **10.68** |

in a significant improvement in performance for all widths, e.g. $11.61\% \rightarrow 8.99\%$. The WCL ensures consistency in the multi-view created by both data augmentation and multiple widths. The Logits Consistency Regularization (LCR) further ensures consistency in the logits dimension, thereby improving performance. These two methods work together and complement each other so that the multiple widths of networks within the slimmable network can interact with each other, leading to better generalization. Adding the ensemble component also resulted in a performance boost, especially for the $1.0\times$ and $0.75\times$ network, e.g. $9.83\% \rightarrow 8.96\%$ for $1.0\times$ network and $10.44\% \rightarrow 9.43\%$ for $0.75\times$ network. Our complete method in the last arrange, which includes all components, achieved the best performance for all widths of network.

**Different Backbones.** In Table 3, we conduct experiments with more backbones on CIFAR10-C to validate the robustness of our method, including MobileNetV1, MobileNetV2, and ViT-Tiny. The results indicate that our method significantly surpasses the current state-of-the-art TTT method

TTAC on a variety of backbones, demonstrating the versatility of SlimTTT. Moreover, our method even achieves competitive performance to TTAC using the $0.5\times$ sub-network, which only contains about $30\%$ parameters compared to the full model.

Table 3: The average error rate ($\%$) results of SlimTTT with different backbones on CIFAR10-C. The numbers presented in red indicate the improvement of our method compared to TTAC.

| Method | Backbone | #Params | Err Avg. | Backbone | #Params | Err Avg. | Backbone | #Params | Err Avg. |
|---|---|---|---|---|---|---|---|---|---|
| TTAC | MobileV1 | 4.2M | 19.80 | MobileV2 | 3.5M | 25.01 | ViT-Tiny | 5.4M | 23.52 |
| SlimTTT | MobileV1 | 4.2M | $\mathbf{16.34}_{(3.46\uparrow)}$ | MobileV2 | 3.5M | $\mathbf{19.09}_{(5.92\uparrow)}$ | ViT-Tiny | 5.4M | $\mathbf{18.87}_{(4.65\uparrow)}$ |
| | MobileV1$_{[0.75\times]}$ | 2.6M | $\mathbf{17.87}_{(1.93\uparrow)}$ | MobileV2$_{[0.75\times]}$ | 2.6M | $\mathbf{21.26}_{(3.75\uparrow)}$ | ViT-Tiny$_{[0.75\times]}$ | 3.2M | $\mathbf{19.35}_{(4.17\uparrow)}$ |
| | MobileV1$_{[0.5\times]}$ | 1.3M | 20.62 | MobileV2$_{[0.5\times]}$ | 2.0M | 25.81 | ViT-Tiny$_{[0.5\times]}$ | 1.6M | $\mathbf{21.77}_{(1.75\uparrow)}$ |
| | MobileV1$_{[0.25\times]}$ | 0.5M | 26.13 | MobileV2$_{[0.25\times]}$ | 1.7M | 31.59 | ViT-Tiny$_{[0.25\times]}$ | 1.2M | 25.08 |

Table 4: The average classification error rates ($\%$) on **CIFAR10-C** dataset using slimmable networks with varying number of widths.

| | $0.25\times$ | $0.375\times$ | $0.5\times$ | $0.625\times$ | $0.75\times$ | $0.875\times$ | $1.0\times$ |
|---|---|---|---|---|---|---|---|
| Individual | - | - | - | - | - | - | 8.95 |
| 2-switch | - | - | 9.72 | - | - | - | 8.85 |
| 4-switch | 10.46 | - | 9.22 | - | 8.85 | - | 8.48 |
| 5-switch | 10.43 | - | 9.17 | 9.06 | 8.72 | - | 8.45 |
| 7-switch | **10.42** | **9.46** | **9.06** | **9.02** | **8.70** | **8.56** | **8.41** |

**Slimmable networks with varying number of widths.** Table 4 presents the results obtained by employing a pre-training model with an increased number of widths. All the results reported in the table were obtained by conducting test-time training and testing on the CIFAR10-C dataset using the same pretrained model with seven sub-networks of varying widths. As shown in Table 4, the performance of each width of network improves as the number of widths increases. This corroborates our claim presented in § 1 that different width networks can capture distinct views of the same input data. Therefore, when the number of different width sub-networks grows, the number of views that can be captured relative to fewer widths also increases. However, when the number of widths increases, the training time at the test-time training phase and the ensemble time in inference phase will also increase. Thus, in practical deployment scenarios, it is essential to consider the trade-off between resources and performance and accordingly determine the number of widths.

**Running cost analysis.** We provide a comprehensive overview on CIFAR10-C of the running costs of our SlimTTT compared to other TTT methods during test-time training phase in Table 5 (batch size=8), including GPU memory, training time per batch, FLOPs and peak memory usage. All results are tested on the actual NVIDIA Jetson TX2 system. As shown in Table 5, the main drawback of our SlimTTT lies in training time. However, our training process does not consume additional device resources, and our SlimTTT method also leads to improved performance across networks of different widths. It is worth noting that during both training and inference phases, SlimTTT offers the flexibility to switch between networks more efficiently compared to traditional methods, without incurring additional overhead.

Table 5: The running cost during test-time training phase of our SlimTTT compared to other TTT methods on the actual NVIDIA Jetson TX2 system.

| Method | Backbone | GPU Memory (G) | Time/Batch (s) | FLOPs (G) | Peak memory (G) |
|---|---|---|---|---|---|
| TTT++ | R-50 | 1.85 | 2.12 | 4.1 | 5.1 |
| TTAC | R-50 | 1.40 | 2.39 | 4.1 | 5.3 |
| SlimTTT | R-50 | 1.11 | 2.14 | 7.8 | 4.2 |
| TTT++ | R-34 | 0.95 | 1.19 | 3.7 | 4.3 |
| TTAC | R-34 | 0.84 | 1.21 | 3.7 | 4.5 |
| SlimTTT | R-50$_{[0.75\times]}$ | 1.04 | 1.74 | 3.7 | 3.6 |
| TTT++ | R-18 | 0.91 | 0.31 | 2.3 | 4.0 |
| TTAC | R-18 | 0.75 | 0.72 | 2.3 | 4.1 |
| SlimTTT | R-50$_{[0.5\times]}$ | 0.92 | 0.80 | 1.4 | 3.0 |

## 5 CONCLUSION AND FUTURE WORKS

In this work, we consider a more practical TTT scenario by incorporating resource constraints. We utilize slimmable networks that can execute various widths of sub-networks and demonstrate that different widths of slimmable networks can capture multi-view input data. To better leverage the multi-view, we introduce WCL, LCR and GFA method to ensure consistency between multi-view in both feature and logits dimensions. Our method provides an effective solution to the practical challenges posed by TTT tasks.

**Future work.** We attempt to incorporate depth refinement on top of SlimTTT to diversify the selection of sub-networks. In our appendix, we add an exit network at the exit of each slimmable ResNet block module to allow us to simultaneously have more sub-networks. We hope that future work can delve deeper into mining components within networks to construct additional sub-networks, thereby leveraging the multi-view capabilities offered by these sub-networks to better address a wider range of resource-efficient challenges.

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
