# OpenReview forum: "Resource Efficient Test-Time Training with Slimmable Network"
_ICLR.cc/2024/Conference — Submitted to ICLR 2024_

### Official Review · Reviewer_aCQv · 2023-10-29

**Soundness:** 4 excellent
**Presentation:** 2 fair
**Contribution:** 3 good
**Rating:** 8
**Confidence:** 3

**Summary:**

This paper introduces SlimTTT, a resource-efficient approach for test-time training. The author proposed a practical scenario of TTT under resource constraints. To address such a challenge, the method utilizes slimmable network that can flexibly switch between different widths of sub-networks for adaptive inference and requires no retraining. The method includes width-enhanced contrastive learning, logit consistency regularization, and global feature alignment to promote representation consistency among sub-networks, in both feature and prediction spaces. The method is demonstrated by the experiment against other TTT methods by the adaption to corrupt samples on several datasets. It can achieve superior performance in the same resource constraint and can be generalized with several different backbones.

**Strengths:**

1. Proposed a new challenge of resource constraint in such a field, which fits in with realistic demands as a result.

2. Proposed a test-time training method for slimmable neural network.

3. Comprehensive experiments that support the methodology and each sub-module of it.

4. The framework considered comprehensively every part of an image classification pipeline.

**Weaknesses:**

1. The definition "Efficient" is vague in the title, that the model result of your method in inference is efficient, but not during the training process（it actually costs more time as claimed in your experiment). Does the efficiency come from utilizing the slimmable models and you proposed a TTT method for them？

2. There are some basic assumptions that should be explained.

3. The purpose of the TTT is to generalize the model when a distribution shift occurs, as your experiment proves by using corrupt samples. This logical correlation can be straightly pointed out.

**Questions:**

1. In some works about explainable networks, the grad-cam result shows what the model is focusing on and indicates the semantical feature that is used to make decisions. Does the bigger-width network contain the attention information of smaller networks well in all these backbones? If not, is the bigger network always better?

2. In some cases of classification, one class can include several visual patterns (eg. some datasets have coarse classes like “vehicle ” including items that do not look exactly the same.) Will these have an impact if you align all networks to the supernet, as it may focus on several instance-level features? I'm also curious if this will have a bigger impact on the contrastive learning that expands the positive data with the different representations from 2 sizes of model.

3. There seems to be a paradox between the purpose of consistency and the ensemble of results in 3.4, if the difference between all subnets to the super net is lowered, will they intend to focus on the same feature? Though it improves the performance, I'm not fully understanding the reason.

---

> ### Author Response · Authors · 2023-11-19
>
> **Q1-1:** The definition "Efficient" is vague in the title, that the model result of your method in inference is efficient, but not during the training process(it actually costs more time as claimed in your experiment).
>
> **A1-1:** Good comment. The term "Efficient" in our paper encompasses two key aspects:
>
> Firstly, efficiency is achieved through the ability to seamlessly switch to smaller and more efficient sub-networks. Despite the test-time training process of our slimTTT taking slightly more time than other methods, we can utilize smaller sub-networks as if they were the largest networks (e.g., employing the R-50 (0.75×) sub-network as the largest network). Remarkably, we observe that the results obtained from the R-50 (0.75×) sub-network demonstrate comparable performance to TTAC (R-50) and even surpass TTT++ (R-50), offering a more efficient alternative.
>
> Secondly, efficiency is also evident in the streamlined approach of deploying and training only one model containing multiple sub-networks for adaptive switching. In contrast, employing previous TTT methods on real world scenarios necessitates deploying and training several models independently to accommodate varying resource constraints. The uploading, downloading, and training processes associated with multiple models of other methods contribute significantly to resource consumption, further underscoring the efficiency of our approach.
>
> **Q1-2:** Does the efficiency come from utilizing the slimmable models and you proposed a TTT method for them？
>
> **A1-2:** We indeed address the resource-efficient test-time training challenges by employing slimmable networks. However, simply incorporating slimmable networks into test-time training poses several challenges: 1. Different widths of networks may interfere with each other, leading to inconsistent update directions, preventing models of various widths from achieving optimal optimization. This issue is more pronounced in test-time training due to the lack of ground truth labels as well as the distribution shift. Therefore, we propose WCL and LCR to solve this challenge, which promotes the consistency between different width of networks in both feature space and prediction space. 2. The GFA is applied to ensure that the test features do not deviate far from the source feature distribution during the whole adaptation process and it can apply a restriction over the multi-view features for each width of sub-networks and ensure the correctness of the multi-view representations.
>
> In the following analytical experiment, we attempt a straightforward application of slimmable networks in test-time training without employing our consistency method (referred to as naive SlimTTT). Unfortunately, this approach yields unsatisfactory results, particularly when applied in an online manner. This is attributed to the fact that our consistency loss functions—WCL, LCR, and GFA—leverage the diverse views captured by sub-networks of varying widths. This facilitates knowledge interaction among slimmable networks, a crucial aspect in the test-time training setting.
>
> | Method | manner | width | Cifar10-C Err. Avg. | manner | width | Cifar10-C Err. Avg. |
> | --- | --- | --- | --- | --- | --- | --- |
> | naive SlimTTT | offline | 1.0 | 10.80 | online | 1.0 | 14.78 |
> |  |  | 0.75 | 11.54 |  | 0.75 | 15.64 |
> |  |  | 0.5 | 12.59 |  | 0.5 | 16.72 |
> |  |  | 0.25 | 15.99 |  | 0.25 | 18.83 |
> | SlimTTT (ours) | offline | 1.0 | **8.33** | online | 1.0 | **10.17** |
> |  |  | 0.75 | **8.63** |  | 0.75 | **10.55** |
> |  |  | 0.5 | **9.12** |  | 0.5 | **11.41** |
> |  |  | 0.25 | **10.68** |  | 0.25 | **14.01** |

---

> > ### Author Response · Authors · 2023-11-19
> >
> > **Q2:** There are some basic assumptions that should be explained.
> >
> > **A2:** Thank you for your suggestion. In the test-time training phase, we follow the approach of TTT++ and TTAC, utilizing an offline training manner. After deploying our slimmable networks to different devices according to their maximum resource constrains, we assume that the test-time training is conducting when the devices have its maximum resources (e.g., the devices are charging).
> >
> > However, motivated by Reviewer 1Qhs, now we find that even though the available compute changes frequently during test-time training in an online manner, our SlimTTT exhibits low sensitivity to changes in resources as shown below. We monitored the sensitivity of the performance in such settings on CIFAR10-C (Gaussian Noise), as different model parameters are updated based on the available compute. We implemented 12 time steps with dynamic resources throughout the period. For example, the resources change after time step 2, so we need to update the 0.75 parameters of the model rather than the whole model.
> >
> > Comparing the performance on dynamically changing resources (the upper half of the table) to that on constant resources (the lower half of the table), the results below reveal that our SlimTTT exhibits low sensitivity to changes in resources. We believe that this dynamic evaluation you mentioned is both important and practical, and we intend to incorporate these findings into our new paper version to encourage future research.
> >
> > |Dynamic resources|Time-step|0|1|2|3|4|5|6|7|8|9|10|11|12|
> > |-|-|-|-|-|-|-|-|-|-|-|-|-|-|-
> > ||FLOPs (G)|4.1|4.1|4.1|2.3|2.3|1.1|1.1|0.3|0.3|1.1|1.1|2.3|2.3
> > ||max width|1|1|1|0.75|0.75|0.5|0.5|0.25|0.25|0.5|0.5|0.75|0.75
> > ||Err. (1.0)|19.53|20.12|17.84|16.31|14.92|14.78|14.73|14.36|13.85|13.55|14.06|13.99|13.82
> > ||Err. (0.75)|22.27|21.29|18.75|17.19|16.02|15.76|15.74|15.33|14.84|14.49|14.95|14.81|14.78
> > ||Err. (0.5)|25.00|23.63|20.57|19.14|17.97|17.51|16.96|16.41|15.89|15.35|15.59|15.43|15.38
> > ||Err. (0.25)|28.13|28.52|25.26|23.93|22.27|21.81|21.04|20.41|19.75|19.18|19.32|18.95|18.96
> > |**Constant resources**|**Time-step**|**0**|**1**|**2**|**3**|**4**|**5**|**6**|**7**|**8**|**9**|**10**|**11**|**12**|
> > ||FLOPs (G)|4.1|4.1|4.1|4.1|4.1|4.1|4.1|4.1|4.1|4.1|4.1|4.1|4.1
> > ||max width |1|1|1|1|1|1|1|1|1|1|1|1|1
> > ||Err. (1.0)|19.53|20.12|17.84|16.41|15.08|14.84|14.73|14.26|13.76|13.36|13.39|13.38|13.37
> > ||Err. (0.75)|22.27|21.29|18.75|17.09|16.02|15.76|15.68|15.14|14.54|14.06|14.24|14.16|14.24
> > ||Err. (0.5)|25.00|23.63|20.57|18.85|17.73|17.51|17.02|16.36|15.76|15.16|15.34|15.10|15.08
> > ||Err. (0.25)|28.13|28.52|25.26|24.22|22.27|21.68|20.87|20.12|19.31|18.79|18.93|18.59|18.51
> >
> > **Q3:** The purpose of the TTT is to generalize the model when a distribution shift occurs, as your experiment proves by using corrupt samples. This logical correlation can be straightly pointed out.
> >
> > **A3:** Thank you for your suggestion. We will add “our conducted experiments use corrupt samples to imitate the distribution shift.” in the new version of our paper.
> >
> > **Q4:** In some works about explainable networks, the grad-cam result shows what the model is focusing on and indicates the semantical feature that is used to make decisions. Does the bigger-width network contain the attention information of smaller networks well in all these backbones? If not, is the bigger network always better?
> >
> > **A4:** Good comment. Bigger-width networks encompass more parameters, potentially containing attention information from smaller networks in certain instances. However, as illustrated in Fig.1 in our introduction, our key observation shows that bigger-width networks can capture entirely distinct perspectives compared to smaller networks. Hence, it is not accurate to claim that larger-width networks always include the attention information of smaller networks.
> >
> > Given that larger-width networks have more parameters, they typically exhibit better overall performance than the smaller networks. Nevertheless, in scenarios where larger-width networks capture entirely different perspectives compared to smaller networks, the smaller networks can complement the knowledge of larger networks. This mutual supplementation leads to direct knowledge exchange and mutual promotion, elucidating why our method proves effective.

---

> > > ### Author Response · Authors · 2023-11-19
> > >
> > > **Q5:** In some cases of classification, one class can include several visual patterns (e.g., some datasets have coarse classes like “vehicle ” including items that do not look exactly the same.) Will these have an impact if you align all networks to the supernet, as it may focus on several instance-level features? I'm also curious if this will have a bigger impact on the contrastive learning that expands the positive data with the different representations from 2 sizes of model.
> > >
> > > **A5:** Thanks for the insightful comment. In our study, a single image (not images belonging to the same class) undergoes two different data augmentations and is then processed by four networks of varying widths, resulting in eight features. These features collectively form the positive data for that particular image. Consequently, the positive data for an image doesn't only involve the largest model size but also encompasses features from other model sizes.
> > >
> > > Furthermore, the multi-views we emphasize in our paper refer to distinct discriminative features that different widths of sub-networks can capture (e.g., the wheel or the car window, both classifiable as a car). Despite datasets having broad classes like "vehicle" that include items with dissimilar appearances, our approach leverages knowledge interaction during test-time training that enables different widths of networks to capture a wide range of views (discriminative features) associated with coarse classes like "vehicle". Therefore, though the fact that the class "vehicle" may encompass items with varying appearances, our networks are capable of capturing the discriminative features specific to each, ultimately leading to enhanced performance compared to individual training.
> > >
> > > Additionally, we conducted an experiment on CIFAR10-C where we don’t align smaller networks to the supernet but to the neighbor width network (e.g., 0.75× width align to the 1.0× width, 0.5× width align to the 0.75× width…). The results indicate that aligning smaller networks to different sizes of the model does not significantly impact contrastive learning.
> > >
> > > | Method | width | Err. Avg. |
> > > | --- | --- | --- |
> > > | neighbor SlimTTT | 1.0 | 8.55 |
> > > |  | 0.75 | 8.94 |
> > > |  | 0.5 | 9.49 |
> > > |  | 0.25 | 11.04 |
> > > | SlimTTT | 1.0 | **8.33** |
> > > |  | 0.75 | **8.63** |
> > > |  | 0.5 | **9.12** |
> > > |  | 0.25 | **10.68** |
> > >
> > > **Q6:** There seems to be a paradox between the purpose of consistency and the ensemble of results in 3.4, if the difference between all subnets to the super net is lowered, will they intend to focus on the same feature? Though it improves the performance, I'm not fully understanding the reason.
> > >
> > > **A6:** Good comment. The primary objective of enforcing consistency is to facilitate knowledge interaction among sub-networks of varying widths. Our aim is to encourage the multi-views captured by different widths of sub-networks to complement each other. Specifically, the supernet guides the smaller networks to capture accurate views for improved image classification, while the smaller networks supplement views that the supernet may have overlooked, enhancing overall robustness.
> > >
> > > However, owing to differences in parameter quantities, networks of various widths cannot capture all perspectives of an object. The supernet tends to cover most views, while the smaller networks retain unique views by aligning not only to the supernet but also to two augmented features of themselves. Consequently, despite that their consistency is promoted, networks of different widths may still capture diverse views. Therefore, we propose an ensemble strategy to guarantee the utilization of all views captured by the networks to improve generalization. This approach has empirically proven to be effective in enhancing the overall performance.

---

> > > > ### Comment · Reviewer_aCQv · 2023-11-23
> > > > **Reply to the Authors' Responses**
> > > >
> > > > Thanks to the authors for your detailed responses. Some of my concerns have been addressed. Therefore, I will raise my score accordingly.

---

> > > > > ### Author Response · Authors · 2023-11-23
> > > > >
> > > > > We thank you for your valuable time and feedback.

---

### Official Review · Reviewer_yJHx · 2023-10-31

**Soundness:** 1 poor
**Presentation:** 3 good
**Contribution:** 2 fair
**Rating:** 5
**Confidence:** 4

**Summary:**

The authors identify two main shortcomings of existing TTT methods: resource inefficiency during deployment across devices and an inability to handle computational budget variations during testing. To address these, the authors propose SlimTTT, a resource-adaptive test-time training framework. They introduce techniques like Width-enhance Contrastive Learning (WCL), Logits Consistency Regularization (LCR), and Global Feature Alignment (GFA) to promote representation consistency. SlimTTT is reported to achieve state-of-the-art results across various adaptation methods and datasets, with a significant reduction in inference complexity. My main concerns lie on the motivation of SlimTTT and its evaluation protocols.

**Strengths:**

The proposed method properly integrates Width-enhance Contrastive Learning, Logits Consistency Regularization, and Global Feature Alignment and thus achieves significant performance improvement compared with prior source-free unsupervised domain adaptation methods, such as SHOT.

The findings of “different width of sub-networks can capture multiple views of the same input data, possessing a talent for enhancing TTT performance” is interesting and provides new insights.

**Weaknesses:**

The motivation for slimmable TTA does not convince me. As all TTA methods are inference-time methods and the adapttion+inference is often conducted on the same device, in Table 1, the computational resource consumption comparison should include both training (GPU memory, wall-clock time) and inference, rather than only #Model Params and Flops during inference. I mean, for a resource-limited device, whether a TTA method can be run is determined by the training resource request, rather than inference.

In Table 1, ResNet-50(1.0$\times$)+TENT+ImageNet-C@56.89 is evaluated in an online manner (to my knowledge). So if SilmTTT is evaluated in an offline manner, the comparison would be unfair.

Does the proposed SlimTTT work well in the online setting? I am curious about the online performance and TTA efficiency (training+inference) comparisons with compared methods.

Comparisons with more resource-efficient TTA methods (perfectly under the online setting) are preferred, such as EATA [Efficient Test-Time Model Adaptation without Forgetting. ICML 2022].

**Questions:**

For all compared methods, are they evaluated in online or offline setting?

---

> ### Author Response · Authors · 2023-11-19
>
> **Q1:** The motivation for slimmable TTA does not convince me. As all TTA methods are inference-time methods and the adapttion+inference is often conducted on the same device, in Table 1, the computational resource consumption comparison should include both training (GPU memory, wall-clock time) and inference, rather than only #Model Params and Flops during inference. I mean, for a resource-limited device, whether a TTA method can be run is determined by the training resource request, rather than inference.
>
> **A1:** Thank you for your suggestion. We follow the offline TTT setting from TTT++ and TTAC. In this offline manner, test-time training and the inference phase are separate and independent. Therefore, in this scenario, it is reasonable to consider the resources during training and inference separately.
>
> The motivation for our SlimTTT actually accounts for both the inference cost and the test-time training cost. As depicted in Fig.1 (left) in our introduction, the blue segment indicates the deployment of networks of various widths after training-time training, where we already take into account the maximum resource constraints of the device during test-time training (e.g., if the maximum resource constraints during test-time training of the device can only support R-50 (0.75×), we deploy the R-50 (0.75×) sub-network utilize it as the largest network). After it meets the resource constraint of each device during test-time training,  SlimTTT has the ability to further switch between sub-networks during test-time inference according to the device’s status, as demonstrated in the orange segment.
>
> To assess the efficacy of our SlimTTT under varying resource constraints during test-time training, we utilize smaller sub-networks as if they were the largest networks (e.g., employing the R-50 (0.75×) sub-network as the largest network). Remarkably, we observe that the results shown in Fig.3 in our experiment chapter obtained from the R-50 (0.75×) sub-network demonstrate comparable performance to TTAC (R-50) and even surpass TTT++ (R-50), offering a more efficient alternative during test-time training phase.
>
> The computational resource consumption comparison and results on ImageNet-C in test-time training phase are shown in the table below.
>
> |Method|Backbone|GPU memory (peak) (G)|wall-clock time (per batch) (s)|FLOPs (G)|Param. (M)|Err. Avg.
> |-|-|-|-|-|-|-
> |TTAC|R-101|>24|2.842|7.8|44.6|43.73
> |TTAC|R-50|23.09|1.885|4.1|25.6|45.71
> |SlimTTT|R-50|22.59|2.253|7.8|25.6|43.98
> |TTAC|R-34|15.98|0.891|3.7|21.8|47.57
> |SlimTTT|R-50 (0.75)|19.81|1.229|3.7|14.7|45.41
> |TTAC|R-18|10.99|0.559|1.8|11.9|51.34
> |SlimTTT|R-50 (0.5)|12.49|0.760|1.3|6.9|48.73
> |SlimTTT|R-50 (0.25)|6.04|0.420|0.2|2.0|58.08
>
> Remarkably, when the available compute changes frequently during test-time training phase, different width of our slimmable network will be updated based on the available compute, while other TTT methods can not conduct test-time training under such dynamic training resource conditions.
>
> We monitored the sensitivity of the performance in such settings on CIFAR10-C (Gaussian_Noise), as different model parameters are updated based on the available compute. We implemented 12 time steps with dynamic resources throughout the period. For example, the resources change after time step 2, so we need to update the 0.75 parameters of the model rather than the whole model.
>
> Comparing the performance on dynamically changing resources (the upper half of the table) to that on constant resources (the lower half of the table), the results below reveal that our SlimTTT exhibits low sensitivity to changes in resources. We believe that this dynamic evaluation you mentioned is both important and practical, and we intend to incorporate these findings into our new paper version to encourage future research.
>
> |Dynamic resources|Time-step|0|1|2|3|4|5|6|7|8|9|10|11|12
> |-|-|-|-|-|-|-|-|-|-|-|-|-|-|-
> ||FLOPs (G)|4.1|4.1|4.1|2.3|2.3|1.1|1.1|0.3|0.3|1.1|1.1|2.3|2.3
> ||max width|1|1|1|0.75|0.75|0.5|0.5|0.25|0.25|0.5|0.5|0.75|0.75
> ||Err. (1.0)|19.53|20.12|17.84|16.31|14.92|14.78|14.73|14.36|13.85|13.55|14.06|13.99|13.82
> ||Err. (0.75)|22.27|21.29|18.75|17.19|16.02|15.76|15.74|15.33|14.84|14.49|14.95|14.81|14.78
> ||Err. (0.5)|25.00|23.63|20.57|19.14|17.97|17.51|16.96|16.41|15.89|15.35|15.59|15.43|15.38
> ||Err. (0.25)|28.13|28.52|25.26|23.93|22.27|21.81|21.04|20.41|19.75|19.18|19.32|18.95|18.96
> |**Constant resources**|**Time-step**|**0**|**1**|**2**|**3**|**4**|**5**|**6**|**7**|**8**|**9**|**10**|**11**|**12**
> ||FLOPs (G)|4.1|4.1|4.1|4.1|4.1|4.1|4.1|4.1|4.1|4.1|4.1|4.1|4.1
> ||max width |1|1|1|1|1|1|1|1|1|1|1|1|1
> ||Err. (1.0)|19.53|20.12|17.84|16.41|15.08|14.84|14.73|14.26|13.76|13.36|13.39|13.38|13.37
> ||Err. (0.75)|22.27|21.29|18.75|17.09|16.02|15.76|15.68|15.14|14.54|14.06|14.24|14.16|14.24
> ||Err. (0.5)|25.00|23.63|20.57|18.85|17.73|17.51|17.02|16.36|15.76|15.16|15.34|15.10|15.08
> ||Err. (0.25)|28.13|28.52|25.26|24.22|22.27|21.68|20.87|20.12|19.31|18.79|18.93|18.59|18.51

---

> > ### Author Response · Authors · 2023-11-19
> >
> > **Q2:** In Table 1, ResNet-50(1.0×)+TENT+ImageNet-C@56.89 is evaluated in an online manner (to my knowledge). So if SilmTTT is evaluated in an offline manner, the comparison would be unfair.
> >
> > **A2:** In our paper, all of results of the compared methods are implemented to be evaluated in an offline manner following the TTT++ and TTAC setting, including TENT (TTT++ includes the implementation of TENT for test-time training in offline manner in its source code, we replicated TENT in offline manner base on the source code of TTT++ across different datasets). So, the comparison is fair.
> >
> > **Q3:** Does the proposed SlimTTT work well in the online setting? I am curious about the online performance and TTA efficiency (training+inference) comparisons with compared methods.
> >
> > **A3:** Good comment. As you suggested, here we further compare SlimTTT to the baseline methods in the online setting. We follow the TTAC (YO) online setting to compare our SlimTTT with other methods on CIFAR10-C. From the results below, we find that our approach still managed to surpass other methods, despite removing the queues present in other approaches (TTAC), resulting in a significant reduction in training time, and we will append the online results in our new version of paper.
> >
> > **CIFAR10-C (online performance):**
> >
> > |Method|width|Birt|Contr|Defoc|Elast|Fog|Frost|Gauss|Glass|Impul|Jpeg|Motn|Pixel|Shot|Snow|Zoom|Avg|
> > |-|-|-|-|-|-|-|-|-|-|-|-|-|-|-|-|-|-|
> > |TENT|1.0|8.22|8.07|9.93|18.29|15.65|14.14|16.60|24.10|25.80|13.39|12.34|11.06|14.75|13.87|7.87|14.27|
> > |TTT++ (w/ q)|1.0|7.70|7.91|9.24|17.55|16.39|12.74|15.49|22.57|22.86|13.02|12.52|11.46|14.45|13.90|7.51|13.69|
> > |TTAC (w/ q)|1.0|6.41|8.05|7.85|14.81|10.28|10.51|13.06|18.36|17.35|10.80|8.97|9.34|11.61|10.01|6.68|10.94|
> > |SlimTTT(w/ q) |1.0|5.67|6.24|7.92| 13.02 | 8.88 | 9.64 | 13.24 | 16.06 | 15.18 | 11.12 | 8.89 | 8.51 | 12.19 | 9.44 | 6.49 | **10.17** |
> > || 0.75 | 5.95 | 6.42 | 8.32 | 13.52 | 9.56 | 10.21 | 13.38 | 16.60 | 15.58 | 11.27 | 9.38 | 8.71 | 12.51 | 10.08 | 6.82 | **10.55** |
> > || 0.5 | 6.55 | 7.33 | 9.05 | 14.54 | 10.01 | 11.02 | 14.68 | 17.37 | 17.22 | 12.25 | 10.05 | 9.60 | 13.53 | 10.63 | 7.27 | **11.41** |
> > || 0.25 | 8.24 | 9.36 | 10.79 | 17.24 | 12.20 | 13.59 | 17.81 | 20.69 | 21.83 | 15.14 | 12.24 | 12.12 | 16.26 | 13.50 | 9.17 | **14.01** |
> > |SlimTTT(w/o q) | 1.0 | 5.51 | 6.17 | 7.90 | 13.14 | 9.18 | 9.43 | 12.66 | 16.65 | 17.63 | 10.81 | 9.07 | 8.66 | 11.87 | 9.41 | 5.88 | 10.26|
> > || 0.75 | 5.88 | 6.69 | 7.87 | 13.73 | 9.75 | 9.64 | 13.55 | 17.52 | 18.18 | 11.35 | 9.54 | 9.05 | 12.08 | 10.02 | 6.16 | 10.73 |
> > || 0.5 | 6.62 | 7.52 | 8.75 | 14.41 | 10.61 | 10.66 | 14.86 | 18.87 | 19.87 | 12.74 | 10.65 | 10.07 | 13.28 | 10.80 | 7.01 | 11.78 |
> > || 0.25 | 8.36 | 10.03 | 10.98 | 18.89 | 14.43 | 13.96 | 19.31 | 23.81 | 26.80 | 16.11 | 13.24 | 13.10 | 17.10 | 14.45 | 9.18 | 15.32 |
> >
> > **CIFAR10-C (online efficiency: TENT, TTAC (with queue), SlimTTT (w/o queue)):**
> >
> > || training time (s)/ iter | training GPU memory | inference time (s) / batch | inference GPU memory |
> > |-|-|-|-|-|
> > |TENT | 0.146 | 12.14G | 0.0574 | 3.59G |
> > |TTAC (w/ queue) | 40.089 | 19.80G | 0.0574 | 3.59G |
> > |SlimTTT (w/o queue) | 4.156 | 16.86G | 0.1133 | 3.59G |
> >
> > **Q4:** Comparisons with more resource-efficient TTA methods (perfectly under the online setting) are preferred, such as EATA.
> >
> > **A4:** Thank you for your suggestion. The comparison with EATA under the online setting on ImageNet-C using ResNet50 as the backbone is shown below, the results show that our SlimTTT can outperform EATA in most of the corruption tasks under online setting.
> >
> > |Method | width | Birt | Contr | Defoc | Elast | Fog | Frost | Gauss | Glass | Impul | Jpeg | Motn | Pixel | Shot | Snow | Zoom | Avg |
> > |-|-|-|-|-|-|-|-|-|-|-|-|-|-|-|-|-|-|
> > |TTAC|1.0|**30.36**| 38.84 | 69.06 | 39.67 | 36.01 | 50.20 | 66.18 | 70.17 | 64.36 | 45.59 | 51.77 | 39.72 | 62.43 | 44.56 | 42.80 | 50.11 |
> > |EATA|1.0|32.00|55.70|66.30|42.40|40.10|54.30|65.00| 66.60|63.20|**45.00**|52.90|**39.30**| 63.10 | 48.60 | 47.20 | 52.11 |
> > |SlimTTT|1.0|31.44|**34.15**|**61.18**|**39.32**|**34.11**|**49.53**|**61.37**|**65.08**|**59.95**|45.04|**44.96**|40.75|**58.73**|**42.05**|**41.15**| **47.25** |
> > || 0.75 | 33.03 | 35.85 | 62.29 | 40.89 | 35.61 | 51.18 | 62.55 | 65.98 | 61.22 | 46.25 | 46.46 | 42.07 | 60.03 | 43.78 | 42.63 | **48.65** |
> > ||0.5|36.17 | 39.12 | 64.06 | 43.8 | 38.74 | 54.53 | 64.93 | 68.05 | 63.71 | 48.89 | 49.38 | 44.78 | 62.47 | 47.07 | 45.59 | **51.42** |
> > ||0.25|42.89 | 46.97 | 70.03 | 50.73 | 46.47 | 63.02 | 70.95 | 73.24 | 69.91 | 55.01 | 56.50 | 51.95 | 68.85 | 54.89 | 52.56 | **58.26** |
> >
> > **Q5:** For all compared methods, are they evaluated in online or offline setting?
> >
> > **A5:** All compared methods are evaluated in offline setting follow TTT++ and TTAC to guarantee a fair comparison. It is worth noting that our method can be used in an online manner, even in situations where training resources are dynamically changing, showcasing strong versatility of SlimTTT.

---

### Official Review · Reviewer_1QhS · 2023-11-01

**Soundness:** 3 good
**Presentation:** 3 good
**Contribution:** 3 good
**Rating:** 6
**Confidence:** 4

**Summary:**

The paper presents a varying resource constraint setting for Test-Time training approaches and proposes an architecture based on the Slimmable Network (Yu et al., 2018). The primary assumption made for the targeted varying resource constraints setting is the dynamically changing computational budget (hardware-based computational constraints or dynamic resource allocation to the algorithm) during inference. Slimmable neural network architectures are specifically designed to provide adaptive accuracy-efficiency trade-offs during model inference on various devices with different budgets of computational resources. In this paper, the authors utilize Width-enhanced Contrastive loss and Logit Consistency Regularization for maintaining consistency between the subnetworks in both features and logits space. Further, the paper also introduces an ensembling strategy based on dynamically changing resources to boost the performance of the architecture. The pipeline for the proposed framework can be summarized in 3 modules 1) Source Training Phase: The pretrained backbone is trained on the source dataset with the primary training objective with an additional auxiliary self-supervised learning task. 2) Target Training Phase:  for this phase, the paper proposes maintaining feature consistency between differently augmented views obtained by both slimmable network structure and data augmentation and adds another learning objective of Logits Consistency Regularization where an augmented version of the sample is sent to the largest network for obtaining a pseudo label which is further used to maintain prediction consistency among all the sub-networks. 3) In the last phase, the paper makes use of the predictions available from different slimmable networks to create an ensemble version of the predictions for boosting prediction performance.

The paper provides empirical results on four widely used benchmarks (ImageNet-C, CIFAR10-C, CIFAR10.1, and CIFAR100-C) with different backbones (ResNet, Mobilienet, and ViT) for four settings of switchable widths in Slimmable Network (1.0×, 0.75×, 0.5×, 0.25×). The reported results highlight an improvement over the compared baselines and show the computation cost comparison via reporting the FLOPs with prediction accuracies. The paper also provides ablation experiments for highlighting the impact of various components in the design of the proposed framework.

**Strengths:**

* The paper highlights an important issue of dynamically changing resource constraints in test-time training settings for deploying models in the real world. The paper utilizes the architecture of slimmable networks to address this issue, making test-time training approaches to incorporate adaptive accuracy-efficiency trade-offs during model inference on various devices with different budgets of computational resources.

* The paper proposes the WCL, LCR, and GFA for exploiting the slimmable network to ensure consistency between multiple views captured by the architecture. The proposed design choices are sufficiently backed up by suitable ablation experiments. Moreover, as an additional advantage, the paper reports the slimmable network for TTT to be effective when compared with other baselines on the same computation budget.

* The paper presents a detailed empirical study with various backbones (3 backbones) and datasets (4 benchmarks) along with different settings of Switches in Slimmable Network (4 in number), with required performance comparison over the computation cost for fair comparison making the results more reliable.

**Weaknesses:**

* One of the primary claims of the paper is the varying inference budget during the inference (also highlighted as Challenge II in Figure 1). Since the paper targets a practical setting of dynamic resource allocation during inference, it is imperative to consider experiments where the available compute changes frequently during inference (test time training). It would be interesting to monitor the sensitivity of the performance in those settings since different parameters of the model will be updated based on the available compute. Low-performance sensitivity on dynamically changing resources will make the method more reliable for practical use cases. Moreover, it’ll promote future research to address the challenges faced in such a setting.

* The proposed framework highly depends on the source domain training however, the training cost and the convergence rate comparison for various architecture is missing from the paper. It would be good to add a comparison of training times/ convergence rates of the proposed architecture to make the approach more transparent for real-world deployment use cases.

**Questions:**

Minor suggestions:
* It would be better to update the caption of Table 3 to highlight the explanation of numbers presented in red for easier readability.
* While making the comparison with existing methods, it would be good to highlight the dependency on the availability of the source dataset. Adding another column for a clear distinction between TTA and TTT approaches will make the paper more transparent.

---

> ### Author Response · Authors · 2023-11-19
>
> **Q1:** It would be interesting to monitor the sensitivity of the performance in those settings since different parameters of the model will be updated based on the available compute. Low-performance sensitivity on dynamically changing resources will make the method more reliable for practical use cases. Moreover, it’ll promote future research to address the challenges faced in such a setting.
>
> **A1:** Good comment. As your suggestion, we conducted experiments on CIFAR10-C (Gaussian_Noise) and CIFAR100-C (Gaussian_Noise) wherein the available compute (e.g., FLOPs) varies frequently during the test-time training phase. In this experiment, we monitored the sensitivity of the performance in such settings, as different model parameters are updated based on the available compute. Specifically, our evaluation spans over 12 time-steps with dynamic resources throughout the period. For example, the resources change after time-step 2 (from 4.1G to 2.3G), so we need to update the 0.75 parameters of the model rather than the whole model.
>
> Comparing the performance on dynamically changing resources (the upper half of the table) to that on constant resources (the lower half of the table), the results below surprisingly reveal that our SlimTTT exhibits low sensitivity to changes in resources (i.e., as of time step 12, the performance of the network trained with dynamic resources is comparable to the performance of the network trained with constant resources), while other TTT methods can not conduct test-time training under such dynamic training resource conditions. We believe that this dynamic evaluation you mentioned is both important and practical, and we intend to incorporate these findings into our new paper version to encourage future research.
>
> **CIFAR10-C (Gaussian_Noise):**
>
> |Dynamic resources|Time-step|0|1| 2 | 3 | 4 | 5 | 6 | 7 | 8 | 9 | 10 | 11 | 12 |
> |-|-|-|-|-|-|-|-|-|-|-|-|-|-|-|
> ||FLOPs | 4.1G | 4.1G | 4.1G | 2.3G | 2.3G | 1.1G | 1.1G | 0.3G | 0.3G | 1.1G | 1.1G | 2.3G | 2.3G |
> ||max width | 1 | 1 | 1 | 0.75 | 0.75 | 0.5 | 0.5 | 0.25 | 0.25 | 0.5 | 0.5 | 0.75 | 0.75 |
> ||Err. (1.0) | 19.53 | 20.12 | 17.84 | 16.31 | 14.92 | 14.78 | 14.73 | 14.36 | 13.85 | 13.55 | 14.06 | 13.99 | 13.82 |
> ||Err. (0.75) | 22.27 | 21.29 | 18.75 | 17.19 | 16.02 | 15.76 | 15.74 | 15.33 | 14.84 | 14.49 | 14.95 | 14.81 | 14.78 |
> ||Err. (0.5) | 25.00 | 23.63 | 20.57 | 19.14 | 17.97 | 17.51 | 16.96 | 16.41 | 15.89 | 15.35 | 15.59 | 15.43 | 15.38 |
> ||Err. (0.25) | 28.13 | 28.52 | 25.26 | 23.93 | 22.27 | 21.81 | 21.04 | 20.41 | 19.75 | 19.18 | 19.32 | 18.95 | 18.96 |
> |**Constant resources** | **Time-step** | **0** | **1** | **2** | **3** | **4** | **5** | **6** | **7** | **8** | **9** | **10** | **11** | **12** |
> ||FLOPs | 4.1G | 4.1G | 4.1G | 4.1G | 4.1G | 4.1G | 4.1G | 4.1G | 4.1G | 4.1G | 4.1G | 4.1G | 4.1G |
> ||max width | 1 | 1 | 1 | 1 | 1 | 1 | 1 | 1 | 1 | 1 | 1 | 1 | 1 |
> ||Err. (1.0) | 19.53 | 20.12 | 17.84 | 16.41 | 15.08 | 14.84 | 14.73 | 14.26 | 13.76 | 13.36 | 13.39 | 13.38 | 13.37 |
> ||Err. (0.75) | 22.27 | 21.29 | 18.75 | 17.09 | 16.02 | 15.76 | 15.68 | 15.14 | 14.54 | 14.06 | 14.24 | 14.16 | 14.24 |
> ||Err. (0.5) | 25.00 | 23.63 | 20.57 | 18.85 | 17.73 | 17.51 | 17.02 | 16.36 | 15.76 | 15.16 | 15.34 | 15.1 | 15.08 |
> ||Err. (0.25) | 28.13 | 28.52 | 25.26 | 24.22 | 22.27 | 21.68 | 20.87 | 20.12 | 19.31 | 18.79 | 18.93 | 18.59 | 18.51 |
>
> **CIFAR100-C (Gaussian_Noise):**
>
> |Dynamic resources|Time-step| 0 | 1 | 2 | 3 | 4 | 5 | 6 | 7 | 8 | 9 | 10 | 11 | 12 |
> |-|-|-|-|-|-|-|-|-|-|-|-|-|-|-|
> ||FLOPs | 2.3G | 2.3G | 2.3G | 4.1G | 4.1G | 0.3G | 0.3G | 1.1G | 1.1G | 2.3G | 2.3G | 1.1G | 1.1G |
> ||max width | 0.75 | 0.75 | 0.75 | 1 | 1 | 0.25 | 0.25 | 0.5 | 0.5 | 0.75 | 0.75 | 0.5 | 0.5 |
> ||Err. (1.0) | 64.06 | 60.94 | 59.90 | 57.81 | 55.47 | 53.52 | 51.79 | 51.56 | 50.95 | 49.84 | 49.64 | 48.76 | 48.62 |
> ||Err. (0.75) | 62.50 | 58.59 | 55.73 | 55.47 | 53.75 | 52.34 | 50.45 | 50.29 | 49.65 | 48.59 | 48.22 | 47.27 | 47.12 |
> ||Err. (0.5) | 63.28 | 59.77 | 58.33 | 58.40 | 56.56 | 54.82 | 52.34 | 52.25 | 51.22 | 50.39 | 50.50 | 49.54 | 49.70 |
> ||Err. (0.25) | 75.78 | 73.83 | 72.14 | 69.14 | 66.88 | 65.23 | 62.61 | 61.91 | 60.94 | 59.77 | 59.73 | 58.85 | 59.07 |
> |**Constant resources** | **Time-step** | **0** | **1** | **2** | **3** | **4** | **5** | **6** | **7** | **8** | **9** | **10** | **11** | **12** |
> ||FLOPs | 4.1G | 4.1G | 4.1G | 4.1G | 4.1G | 4.1G | 4.1G | 4.1G | 4.1G | 4.1G | 4.1G | 4.1G | 4.1G |
> ||max width | 1 | 1 | 1 | 1 | 1 | 1 | 1 | 1 | 1 | 1 | 1 | 1 | 1 |
> ||Err. (1.0) | 62.50 | 58.2 | 56.51 | 54.69 | 52.50 | 50.78 | 48.66 | 48.14 | 47.31 | 46.56 | 46.09 | 45.31 | 45.13 |
> ||Err. (0.75) | 62.50 | 58.59 | 55.73 | 55.47 | 53.75 | 52.21 | 50.22 | 49.71 | 48.96 | 48.12 | 47.73 | 46.81 | 46.57 |
> ||Err. (0.5) | 63.28 | 59.77 | 58.33 | 58.40 | 56.56 | 54.82 | 52.23 | 51.76 | 50.52 | 49.45 | 49.43 | 48.50 | 48.62 |
> ||Err. (0.25) | 75.78 | 73.83 | 72.14 | 69.14 | 66.88 | 65.23 | 62.72 | 62.01 | 61.02 | 59.84 | 59.87 | 59.05 | 59.25 |

---

> > ### Author Response · Authors · 2023-11-19
> >
> > **Q2:** The proposed framework highly depends on the source domain training however, the training cost and the convergence rate comparison for various architecture is missing from the paper. It would be good to add a comparison of training times/ convergence rates of the proposed architecture to make the approach more transparent for real-world deployment use cases.
> >
> > **A2:** Thank you for your suggestions. We provide the results below to show the training times of the proposed architecture (ResNet50, MobileNetV1, MobileNetV2 and ViT-Tiny) and we provide the convergence rate comparison (2000 epochs) for various architecture (ResNet50, MobileNetV1, MobileNetV2) in our new version of supplementary (highlighted in blue).
> >
> > Training Times per epoch:
> >
> > | Backbone | Training Time/epoch |
> > | --- | --- |
> > | ResNet50 | 173.48s |
> > | MobileNetV1 | 28.31s |
> > | MobileNetV2 | 43.07s |
> > | ViT-Tiny | 202.7s |
> >
> > **S1:** It would be better to update the caption of Table 3 to highlight the explanation of numbers presented in red for easier readability.
> >
> > **A1:** Thank you for your suggestion. We have modified the caption of Table 3 to “The average error rate (%) results of SlimTTT with different backbones on CIFAR10-C. The numbers presented in red indicate the improvement of our method compared to TTAC”, and have updated it in our paper (highlighted in blue).
> >
> > **S2:** While making the comparison with existing methods, it would be good to highlight the dependency on the availability of the source dataset. Adding another column for a clear distinction between TTA and TTT approaches will make the paper more transparent.
> >
> > **A2:** Thank you for your suggestion. We will add another column like we show below to highlight the dependency on the availability of the source dataset in Table1 in the future version. “SSL head” in the table means whether we need to train an auxiliary branch in source training phase, “Availability of the source” means whether we can access the source data during test-time training.
> >
> > |  | BN (TTA) | TTT-R (TTT) | SHOT (Source-free) | TENT (TTA) | TTT++ (TTT) | TTAC (TTT) | SlimTTT (TTT) |
> > | --- | --- | --- | --- | --- | --- | --- | --- |
> > | SSL head | × | √ | × | × | √ | √ | √ |
> > | Availability of the source data | × | × | × | × | × | × | × |

---

> > > ### Comment · Reviewer_1QhS · 2023-11-22
> > >
> > > Thank you for sharing the results. The additional details, like the requirement for retraining of SSL on source dataset, help improve the paper's clarity.

---

> > ### Comment · Reviewer_1QhS · 2023-11-22
> >
> > Thank you for adding the additional results.
> >
> > The results of dynamically changing computational budget do reveal a low sensitivity to the dynamic changes for CIFAR10C, and it is somewhat surprising that for a few cases, the model performs better over low dynamic resources. As a suggestion, an average over multiple runs column would make it easier for comparison.
> >
> > CIFAR10C
> >
> > | Constant Resource | Dynamic Resource | Error Difference (Constant - Dynamic) |
> > |-------------------|------------------|--------------------------------------|
> > | 15.39             | 15.52769231      | -0.1376923077                       |
> > | 16.40307692      | 16.63230769      | -0.2292307692                       |
> > | 17.93153846      | 18.06384615      | -0.1323076923                       |
> > | 21.93846154      | 22.11769231      | -0.1792307692                       |
> >
> > CIFAR100C
> > | Constant Resource | Dynamic Resource | Error Difference (Constant - Dynamic) |
> > |-------------------|------------------|--------------------------------------|
> > | 50.95230769      | 54.06615385      | -3.113846154                        |
> > | 52.02846154      | 52.30538462      | -0.2769230769                       |
> > | 53.97461538      | 54.39230769      | -0.4176923077                       |
> > | 65.13538462      | 65.06769231      | 0.06769230769                       |
> >
> >
> > For CIFAR100C, the average performance difference seems to be significant. A brief discussion regarding the observation would also be helpful if there are cases where, on a longer range of dynamic resource changes, the model collapses due to massive error accumulation over time. Mentioning these observations clearly will improve the transparency of the work.

---

> > > ### Author Response · Authors · 2023-11-22
> > >
> > > **Q:** The results of dynamically changing computational budget do reveal a low sensitivity to the dynamic changes for CIFAR10C, and it is somewhat surprising that for a few cases, the model performs better over low dynamic resources. As a suggestion, an average over multiple runs column would make it easier for comparison. For CIFAR100C, the average performance difference seems to be significant. A brief discussion regarding the observation would also be helpful if there are cases where, on a longer range of dynamic resource changes, the model collapses due to massive error accumulation over time. Mentioning these observations clearly will improve the transparency of the work.
> > >
> > > **A:**
> > >
> > > Thank you very much for your suggestion! The results we previously reported at each time-step were already averages across all previous steps. Namely, the last column in table shows the average error rates over twelve time-steps and you can calculate the difference between constant and dynamic cases directly from it. For a more comprehensive results, as you suggested, we have now adjusted the table presentation and displayed the instantaneous results on CIFAR10-C (Guassian_Noise) and CIAFR100-C (Guassian_Noise) at each time-step on a longer range of dynamic resource changes. We report their averages across the whole process at the last column, where the values in parentheses represent the error difference (constant - dynamic).
> > >
> > > **CIFAR10-C (batchsize=256, total steps=40):**
> > >
> > > | Dynamic resources | Time-step | 1 | 2 | … | 12 | 13 | … | 20 | 21 | … | 38 | 39 | 40 | Avg. 40 steps |
> > > | --- | --- | --- | --- | --- | --- | --- | --- | --- | --- | --- | --- | --- | --- | --- |
> > > |  | FLOPs | 2.3G | 2.3G |  | 1.1G | 1.1G |  | 0.3G | 0.3G |  | 2.3G | 2.3G | 2.3G |  |
> > > |  | max width | 0.75 | 0.75 |  | 0.5 | 0.5 |  | 0.25 | 0.25 |  | 0.75 | 0.75 | 0.75 |  |
> > > |  | Err. (1.0) | 21.88 | 21.88 |  | 13.28 | 11.71 |  | 12.89 | 12.89 |  | 16.41 | 13.28 | 13.67 | 12.96 |
> > > |  | Err. (0.75) | 22.65 | 20.71 |  | 13.28 | 15.63 |  | 12.50 | 12.50 |  | 17.58 | 12.50 | 14.45 | 13.73 |
> > > |  | Err. (0.5) | 25.00 | 22.66 |  | 13.28 | 15.23 |  | 13.28 | 12.11 |  | 17.97 | 15.63 | 13.67 | 14.63 |
> > > |  | Err. (0.25) | 28.52 | 29.30 |  | 14.84 | 17.97 |  | 17.97 | 14.45 |  | 20.70 | 18.36 | 15.63 | 17.80 |
> > > | **Constant resources** | **Time-step** | **1** | **2** | **…** | **12** | **13** | **…** | **20** | **21** | **…** | **38** | **39** | **40** | **Avg. 40 steps (error difference)** |
> > > |  | FLOPs | 4.1G | 4.1G |  | 4.1G | 4.1G |  | 4.1G | 4.1G |  | 4.1G | 4.1G | 4.1G |  |
> > > |  | max width | 1 | 1 |  | 1 | 1 |  | 1 | 1 |  | 1 | 1 | 1 |  |
> > > |  | Err. (1.0) | 19.53 | 20.71 |  | 12.89 | 13.28 |  | 12.89 | 12.11 |  | 16.41 | 13.67 | 13.67 | 12.81 (-0.15) |
> > > |  | Err. (0.75) | 22.27 | 20.31 |  | 13.28 | 14.84 |  | 12.11 | 12.50 |  | 15.63 | 13.28 | 15.23 | 13.51 (-0.22) |
> > > |  | Err. (0.5) | 25.00 | 22.27 |  | 12.50 | 14.84 |  | 13.67 | 11.72 |  | 17.58 | 14.06 | 15.23 | 14.55 (-0.08) |
> > > |  | Err. (0.25) | 28.13 | 28.91 |  | 15.23 | 17.58 |  | 17.19 | 14.45 |  | 19.92 | 18.36 | 17.58 | 17.62 (-0.18) |
> > >
> > > **CIFAR100-C (batchsize=128, total steps=80):**
> > >
> > > | Dynamic resources | Time-step | 1 | 2 | … | 12 | 13 | … | 20 | 21 | … | 78 | 79 | 80 | Avg. 80 steps |
> > > | --- | --- | --- | --- | --- | --- | --- | --- | --- | --- | --- | --- | --- | --- | --- |
> > > |  | FLOPs | 2.3G | 2.3G |  | 1.1G | 1.1G |  | 0.3G | 0.3G |  | 2.3G | 2.3G | 2.3G |  |
> > > |  | max width | 0.75 | 0.75 |  | 0.5 | 0.5 |  | 0.25 | 0.25 |  | 0.75 | 0.75 | 0.75 |  |
> > > |  | Err. (1.0) | 64.06 | 57.81 |  | 39.06 | 46.88 |  | 38.28 | 39.84 |  | 40.63 | 49.22 | 37.50 | 41.08 |
> > > |  | Err. (0.75) | 62.50 | 54.69 |  | 36.72 | 45.31 |  | 44.53 | 41.41 |  | 36.72 | 51.56 | 39.06 | 41.96 |
> > > |  | Err. (0.5) | 63.28 | 56.25 |  | 39.06 | 51.56 |  | 51.56 | 41.41 |  | 37.50 | 49.22 | 41.41 | 45.15 |
> > > |  | Err. (0.25) | 75.78 | 71.88 |  | 49.22 | 61.72 |  | 56.25 | 53.13 |  | 46.88 | 55.47 | 56.25 | 53.57 |
> > > | **Constant resources** | **Time-step** | **1** | **2** | **…** | **12** | **13** | **…** | **20** | **21** | **…** | **78** | **79** | **80** | **Avg. 80 steps (error difference)** |
> > > |  | FLOPs | 4.1G | 4.1G |  | 4.1G | 4.1G |  | 4.1G | 4.1G |  | 4.1G | 4.1G | 4.1G |  |
> > > |  | max width | 1 | 1 |  | 1 | 1 |  | 1 | 1 |  | 1 | 1 | 1 |  |
> > > |  | Err. (1.0) | 62.50 | 53.91 |  | 36.72 | 42.97 |  | 37.50 | 36.72 |  | 39.84 | 50.78 | 35.94 | 39.94 (-1.14) |
> > > |  | Err. (0.75) | 62.50 | 54.69 |  | 36.72 | 42.97 |  | 45.31 | 40.63 |  | 36.72 | 50.78 | 38.28 | 41.63 (-0.33) |
> > > |  | Err. (0.5) | 63.28 | 56.25 |  | 38.28 | 50.00 |  | 52.34 | 41.41 |  | 38.28 | 49.22 | 41.41 | 44.97 (-0.18) |
> > > |  | Err. (0.25) | 75.78 | 71.88 |  | 50.00 | 61.72 |  | 55.47 | 53.91 |  | 46.88 | 55.47 | 56.25 | 53.71 (+0.14) |

---

> > > > ### Author Response · Authors · 2023-11-22
> > > >
> > > > **A:** Based on the average results of networks with various widths trained for an extended period, we observed that the average performance of networks under dynamic resource conditions is inferior to that under static resource conditions with a small gap. Our updated supplementary material provides a comparison figure (highlighted in blue) of the convergence rates for the supernet under dynamic and static resource conditions. The figure shows that as training time increases, the average performance of networks under dynamic resource conditions gradually converges with that under static resource conditions.
> > > >
> > > > According to this observation, we conclude that the optimal performance achievable by networks under dynamic resource conditions is not as high as that under static resource conditions. Additionally, the initial convergence rate of the networks under dynamic resource conditions is slower, explaining the significant performance difference observed after 12 steps in our initial response, which means the dynamic resources condition does have a certain impact on optimization of the slimmable networks. However, with prolonged training, the performance gap diminishes, reaching their respective optima. The results from longer training durations indicate that our approach does not suffer from massive error accumulation that could lead to model crashes. Our intuitive explanation for this phenomenon is that the consistency in multi-view at the feature and prediction space, as proposed in our approach, assists the optimization objectives of each network to gradually become more consistent as training progresses.
> > > >
> > > > Furthermore, from the results of each step in the table, we were pleasantly surprised to find that, at certain steps, the model's performance under dynamic resource changes is better than its performance under static conditions. Our intuitive explanation for this phenomenon is that when adjusting the width based on dynamic resource changes at a particular step, the calculated gradient direction may be more favorable for the update of a specific width network. Consequently, the performance of that width network tends to be better than under static conditions at that step. This inspires our future research on the selection of updating parameters for networks of different widths.

---

### Official Review · Reviewer_by2E · 2023-11-03

**Soundness:** 2 fair
**Presentation:** 2 fair
**Contribution:** 2 fair
**Rating:** 5
**Confidence:** 4

**Summary:**

This paper proposed to leverage slimmable networks in the scenario of test-time training to allow the model to meet different resource budgets during test-time training. The paper proposed width-enhanced contrastive learning which is to conduct contrastive learning among different network width to learn different representations. The proposed method shows better performance than previous	works at different resource budgets

**Strengths:**

1.	The paper has a clear explanation about the background and objectives of this work.
2.	The proposed method shows better performance than previous works at different resource budgets.

**Weaknesses:**

1.	It seems that this work is pretty much slimmable network, but just in the setting of test-time training. I don’t see what are the unique challenges in applying slimmable networks in the test-time training? The test-time training seems to be the same as training-time training, but just without labels. Then the problem seems to be how to apply slimmable networks in the un-supervised setting, which has been studied in previous works [1].
2.	The motivation that different sub-networks could capture different image features has been studied in [2].
3.	In Table 1, the other methods should also use ResNet-50 with different widths to have a fair comparison.
4.	This work finetuned the pre-trained slimmable networks on ImageNet and ImageNet-C, what about other works?
5.	In Table 5, why is the training cost is comparable or even faster than TTT++ and TTAC? The proposed method needs to forward and backward multiple times, I am assuming it should be more training expensive.


[1] Slimmable Networks for Contrastive Self-supervised Learning.
[2] MutualNet: Adaptive ConvNet via Mutual Learning from Different Model Configurations. TPAMI.

**Questions:**

Please see the weakness part

---

> ### Author Response · Authors · 2023-11-18
>
> **Q1-1:** The test-time training seems to be the same as training-time training, but just without labels.
>
> **A1-1:** Good comment. Test-time training (TTT) improves the model’s generalization on specific test data distribution in a way that is beyond the capabilities of training-time training. Since the distribution of the test data can be diverse and different from training data distribution, it is highly difficult for training-time training alone to train a model that can generalize to all possible test data distribution. TTT, on the other hand, provides a solution to enhance the model’s generalization on specific test data distribution. Particularly, TTT takes a training-time trained model, and then updates the model using unlabeled test data to directly adapt it towards the test distribution. The adapted model is finally used for inference. Therefore, TTT is quite different to training-time training, regardless of whether it has training labels or not.
>
> **Q1-2:** Then the problem seems to be how to apply slimmable networks in the un-supervised setting, which has been studied in previous works [1].
>
> **A1-2:** In spite of both using unlabeled data, TTT and unsupervised learning are quite different. The goal of TTT is to improve the supervised learning model’s generalization performance on test data by directly adapting the model towards them.  By contrast, unsupervised learning aims to learn robust representation that serves as a decent starting point for downstream tasks. Beyond this fundamental discrepancy, SlimTTT differs from [1] in methods. Firstly, there are distinctions between our contrastive learning approach and [1]. SlimTTT aggregates the multi-views captured by networks of varying widths and employs a multi-positive NCE loss, whereas [1] follows the standard SimCLR-MoCo paradigm to train each sub-networks. Our SlimTTT better facilitates knowledge interaction among networks of different widths. Secondly, we employ LCR and GFA to additionally promote the consistency of these multi-views, enhancing the networks' ability to handle distribution shifts.
>
> To further demonstrate the superiority of TTT compared to unsupervised/self-supervised pretraining in the scenario of adapting to specific test distribution, we consider the following alternative involving [1]. We replace our supervised training-time training on CIFAR10 (C10) by unsupervised learning on both C10 and C10-C (Gaussian Noise (G_N)) using [1] and then finetune on C10. We do not conduct test-time training for this alternative and directly test on C10-C (G_N). Nevertheless, the results reveal that the method proposed in [1] struggles to handle distribution shifts in this process, leading to subpar performance on C10-C (G_N). Additionally, it's worth noting that our SlimTTT is adaptable to an online TTT setting, a capability not present in the approach proposed in [1].
>
> |Method|Protocol|Training-time training|Test-time training|Inference|Width|Err. (G_N)|
> |-|-|-|-|-|-|-
> |[1]|Self-supervised|unsupervised|None|on C10-C|1.0|16.06
> |||on C10 + C10-C|||0.75|15.44
> |||then supervised|||0.5|15.69
> |||on C10|||0.25|16.91
> |SlimTTT|Test-time training|supervised|unsupervised|on C10-C|1.0| **10.41**
> |||on C10|on C10-C||0.75| **11.04**
> ||||||0.5| **11.80**
> ||||||0.25| **13.96**
>
> **Q1-3:** what are the unique challenges in applying slimmable networks in the test-time training?
>
> **A1-3:** Simply incorporating slimmable networks into TTT poses several challenges: 1. Different widths of networks may interfere with each other, leading to inconsistent update directions, preventing models from achieving optimal optimization. This issue is more pronounced in TTT due to the lack of labels as well as the distribution shift. Therefore, we propose WCL and LCR to solve this challenge, which promotes the consistency between different width of networks. 2. The GFA is applied to ensure that the test features do not deviate far from the source feature distribution and it can apply a restriction over the multi-view features for each width of networks.
>
> In the following analytical experiment on C10-C, we attempt a straightforward application of slimmable networks in test-time training without employing our consistency method (referred to as naive SlimTTT). Unfortunately, this approach yields unsatisfactory results, particularly when applied in an online manner. This is attributed to the fact that our consistency loss functions—WCL, LCR, and GFA—leverage the diverse views captured by networks of varying widths. This facilitates knowledge interaction among slimmable networks, a crucial aspect in the test-time training setting.
>
> |Method|width|manner|Err. Avg.|manner|Err. Avg.
> |-|-|-|-|-|-
> |naive SlimTTT|1.0|offline|10.80|online|14.78
> ||0.75||11.54||15.64
> ||0.5||12.59||16.72
> ||0.25||15.99||18.83
> |SlimTTT|1.0|offline|**8.33**|online|**10.17**
> ||0.75||**8.63**||**10.55**
> ||0.5||**9.12**||**11.41**
> ||0.25||**10.68**||**14.01**

---

> > ### Author Response · Authors · 2023-11-18
> >
> > **Q2:** The motivation that different sub-networks could capture different image features has been studied in [2].
> >
> > **A2:** Thank you for bringing this to our attention. MutualNet [2] observes that distinct model-resolution configurations can capture varying image features. However, it's essential to note that the observations and the underlying motivation differ between MutualNet and our approach.
> >
> > MutualNet claims that due to the shared parameters, the sub-networks can learn multi-scale representations, in terms of pairing images of different resolutions with networks of various widths. They supervise the sub-networks by the supernet prediction with largest image resolution. Thus, the knowledge transfer direction is only from supernet to the sub-networks. Unlike MutualNet, our SlimTTT proposes consistency loss function WCL to explicitly combine supernet and other sub-networks to do contrastive learning in feature space, which enables explicit knowledge transfer for both direction.
> >
> > Moreover, MutualNet identifies that distinct model-resolution configurations, such as 1.0×-224 and 0.75×-128, can emphasize different image features. Specifically, larger networks with higher resolutions tend to focus more on details, while smaller networks with lower resolutions prioritize global structures. It's noteworthy that the image features captured by various networks are influenced by the resolution of the image itself, while the multi-view in our SlimTTT rely solely on the width of sub-networks.
> >
> > Furthermore, the concept of multi-view captured by our SlimTTT differs from the image features identified by MutualNet. Drawing on the principles outlined in [3], we demonstrate that sub-networks of varying widths can capture distinct discriminative multi-views of an object, which is supported by visualization experiments and a theoretical toy example. For instance, certain widths of sub-networks might focus on the head of a dog, while others capture the tail of the same dog. These discriminative features are entirely distinct and can both contribute effectively to the classification of a dog. These multi-views are different from the detailed or global features proposed by MutualNet, which hinge on the resolution of the image.
> >
> > Finally, we conducted the following experiment to demonstrate that the distinct image features captured by MutualNet using different image resolutions are orthogonal to the multi-views we identified. We treat images of different resolutions as diverse forms of data augmentation. In contrast to MutualNet randomly inputting images of a certain resolution into a network of a particular width, each width of network can observe images of all resolutions once a time in our method. This approach increases the views we claimed in our paper, and through our designed WCL and LCR, we promote consistency between these views. As shown in the table below, we achieve better results on ImageNet-C in online manner after adding images of different resolutions. This indicates that MutualNet and our SlimTTT are orthogonal. It's important to note that these results are preliminary, and we plan to delve deeper into the benefits that images of different resolutions can bring to our method in the future.
> >
> > | Method | width | Gaussian_Noise | Snow | Brightness |
> > | --- | --- | --- | --- | --- |
> > | SlimTTT | 1.0 | 61.37 | 42.05 | 31.44 |
> > |  | 0.75 | 62.55 | 43.78 | 33.03 |
> > |  | 0.5 | 64.93 | 47.07 | 36.17 |
> > |  | 0.25 | 70.95 | 54.89 | 42.89 |
> > | SlimTTT + resolution | 1.0 | 61.17 | 41.74 | 31.50 |
> > |  | 0.75 | 63.27 | 43.03 | 32.77 |
> > |  | 0.5 | 64.52 | 49.03 | 35.83 |
> > |  | 0.25 | 70.05 | 54.37 | 42.77 |
> >
> > Therefore, both the concept of multi-views and the rationale for their inclusion in our SlimTTT differ from the approach taken by MutualNet.
> >
> > [3] Towards Understanding Ensemble, Knowledge Distillation and Self-Distillation in Deep Learning.
> >
> > **Q3:** In Table 1, the other methods should also use ResNet-50 with different widths to have a fair comparison.
> >
> > **A3:** Thank you for your suggestions. We conducted a comparison of our method with other TTT methods using ResNet-50 with varying widths on Cifar10-C and Cifar100-C, and the results are presented in the table below. The findings demonstrate that our method consistently outperforms other TTT methods, attributed to the unique loss we designed to guarantee the consistency of networks with different widths.
> >
> > | Method | width | Cifar10-C Err. Avg. | Cifar100-C Err. Avg. |
> > | --- | --- | --- | --- |
> > | TTT++ | 1.0 | 9.68 | 32.71 |
> > | TTAC | 1.0 | 9.24 | 31.60 |
> > | SlimTTT (ours) | 1.0 | **8.33** | **29.36** |
> > | TTT++ | 0.75 | 10.62 | 34.72 |
> > | TTAC | 0.75 | 10.08 | 33.90 |
> > | SlimTTT (ours) | 0.75 | **8.63** | **30.34** |
> > | TTT++ | 0.5 | 11.47 | 36.42 |
> > | TTAC | 0.5 | 11.35 | 35.73 |
> > | SlimTTT (ours) | 0.5 | **9.12** | **32.07** |
> > | TTT++ | 0.25 | 12.88 | 40.32 |
> > | TTAC | 0.25 | 13.46 | 40.06 |
> > | SlimTTT (ours) | 0.25 | **10.68** | **37.71** |

---

> > > ### Author Response · Authors · 2023-11-18
> > >
> > > **Q4:** This work finetuned the pre-trained slimmable networks on ImageNet and ImageNet-C, what about other works?
> > >
> > > **A4:** Thank you for pointing it out. In our study, we initially conduct standard training-time training procedure for slimmable networks on training dataset ImageNet. Subsequently, we update/finetune the pre-trained slimmable network on test dataset ImageNet-C **(without labels)** without accessing ImageNet anymore. The full procedure follows the previous works like TTT++ and TTAC. It's important to note that all the methods compared in our paper follow this standardized process to ensure a fair and equitable comparison. We will rephrase this process in our new version of paper to avoid confusions.
> > >
> > > **Q5:** In Table 5, why is the training cost is comparable or even faster than TTT++ and TTAC? The proposed method needs to forward and backward multiple times, I am assuming it should be more training expensive.
> > >
> > > **A5:** Good comment. In TTT++ and TTAC, maintaining a queue is necessary for projection alignment to update the self-supervised head (SSH). Additionally, TTAC involves classwise alignment and filtering. Although our SlimTTT needs to forward and backward multiple times, our method doesn’t need to use class alignment or ssh alignment like TTT++ and TTAC, thus the total training cost of our SlimTTT can be comparable with TTT++ and TTAC.
> > >
> > > To fairly compare the training cost of our method with TTT++ and TTAC, we conduct experiment on CIFAR10-C (batchsize=8) in NVIDIA Jetson TX2 to veirfy the training cost of different module. The results are shown in the table below, the total training costs of our SlimTTT are comparable with TTT++ and TTAC, which means our SlimTTT is simple but effective.
> > >
> > > |  | ssl_con | pre_reg | class_align | ext_align | ssh_align | optimize | total | Err. Avg. |
> > > | --- | --- | --- | --- | --- | --- | --- | --- | --- |
> > > | TTT++ (ResNet50) | 0.7906s | 0 | 0 | 0.4964s | 0.7467s | 0.0820s | 2.1157s | 9.80 |
> > > | TTAC (ResNet50) | 0.6773s | 0 | 0.6042s | 0.3993s | 0.6173s | 0.0938s | 2.3919s | 8.52 |
> > > | SlimTTT (ResNet50) | 1.1958s | 0.1461s | 0 | 0.6794s | 0 | 0.1207s | 2.1420s | 8.33 |

---

> > ### Comment · Reviewer_by2E · 2023-11-19
> > **Follow up quesstions**
> >
> > Q1-2: I still didn't see the fundamental difference between the proposed method and unsupervised slimmable networks such as [1]. The WCL is combining the multi-positive NCE loss in contrastive learning with slimmable network. LRC is the same as the loss in slimmable network (largest network to supervise sub-networks). GFA is also proposed in previous works. Prediction ensemble is also a widely used tricks for improved performance. And the method is the combination of these techniques. I don't think there are many technical contributions. And for comparison, I think the author could apply some unsupervised slimmable networks works to the test-time training stage in their to see the difference.
> >
> > Another question: Table 2 shows that GFA gives most of the improvements. It is proposed in previous works and not clearly explained how it is applied in the proposed method.

---

> > > ### Author Response · Authors · 2023-11-20
> > >
> > > **Q1:** I still didn't see the fundamental difference between the proposed method and unsupervised slimmable networks such as [1]. The WCL is combining the multi-positive NCE loss in contrastive learning with slimmable network. LRC is the same as the loss in slimmable network (largest network to supervise sub-networks). GFA is also proposed in previous works. Prediction ensemble is also a widely used tricks for improved performance. And the method is the combination of these techniques. I don't think there are many technical contributions. And for comparison, I think the author could apply some unsupervised slimmable networks works to the test-time training stage in their to see the difference.
> > >
> > > **A1:** We will describe the fundamental difference between the proposed method and unsupervised slimmable networks such as [1] from three aspects: 1. the differences between the TTT setting and unsupervised setting, 2. the differences of the techniques proposed in our method and [1], 3. the differences of the motivation of our method and unsupervised method as [1].
> > >
> > > 1. **The differences of the TTT setting and unsupervised learning setting.**
> > >
> > > In unsupervised learning, methods first utilize unlabeled training data to train the feature encoder. To perform classification, they subsequently learn a classifier in the supervised manner (usually with the feature encoder fixed) for downstream tasks. In contrast, our approach in the TTT setting trains the feature encoder, the classifier and an auxiliary projection head,  (altogether referred to as the Y-structure in TTT) using labeled training data in a supervised manner. Then at test time, we fix the classifier while improving the feature encoder in a self-supervised manner. Such fundamental procedure difference prevents the direct application of unsupervised slimmable network methods in test-time training.
> > >
> > > SlimTTT builds upon and improves the Y-structure in TTT. We update the feature encoder through the self-supervised branch, mitigating the impact of distribution shift. This allows the slimmable network to robustly adapt using unlabeled test data with distribution shift. Additionally, we update the feature encoder through the main branch (classifier branch) using LCR while keeping the classifier fixed. This approach prevents the classifier from overfitting to the test data.
> > >
> > > Since that the unsupervised slimmable network method cannot be directly applied to TTT setting, methods like [1] have to be modified in the first place. In our initial response, the comparison with [1] illustrated the superiority of TTT over unsupervised learning in terms of generalization performance. Here, we further demonstrate the advantages of our approach over the unsupervised method as presented in [1], when adapted for TTT settings, with a comparative experiment on CIFAR10-C (Gaussian_Noise). The unsupervised techniques in [1] involve contrastive learning and online distillation using KL-Divergence (KLD). From the results we can find that, even with modification, [1] alone cannot address TTT due to the distribution shift. After applying GFA module, the performance of [1] greatly improves, yet our SlimTTT still performs better thus highlighting the overall effectiveness of our approach.
> > >
> > > | Method | Err. (1.0) | Err. (0.75) | Err. (0.5) | Err. (0.25) |
> > > | --- | --- | --- | --- | --- |
> > > | [1] | 23.47 | 24.07 | 24.42 | 27.53 |
> > > | [1] (w/ GFA) | 11.72 | 12.46 | 13.29 | 15.86 |
> > > | SlimTTT | **10.41** | **11.04** | **11.80** | **13.96** |
> > >
> > > Additionally, we conduct ablation experiments to individually analyze the effectiveness of WCL vs. contrastive learning in [1] and LCR vs. KLD in [1]. The results demonstrate that both components of WCL and LCR outperform [1]. Further analysis of the specific advantages of each component will be presented in the next section.
> > >
> > > | Method | Err. (1.0) | Err. (0.75) | Err. (0.5) | Err. (0.25) |
> > > | --- | --- | --- | --- | --- |
> > > | [1] (w/ GFA, w/o KLD) | 11.96 | 12.21 | 13.88 | 16.65 |
> > > | SlimTTT (w/o LCR) | **10.96** | **11.75** | **13.34** | **16.31** |
> > > | [1] (w/ GFA, w/o contrastive) | 12.58 | 13.41 | 14.77 | 16.11 |
> > > | SlimTTT (w/o WCL) | **11.03** | **11.38** | **12.47** | **15.23** |

---

> > > > ### Author Response · Authors · 2023-11-20
> > > >
> > > > 2. **The differences between the techniques proposed in our method and in related methods.**
> > > >
> > > > Firstly, the WCL is different from the existing contrastive learning methods that are leveraged to training a slimmable network. It is a modification more suitable for slimmable network in TTT setting. First of all, **WCL does not require a queue**, whereas the traditional contrastive loss like SimCLR-MoCo used in [1] relies on a queue, incurring additional overhead, making it unsuitable for the TTT setting. **Moreover, in the construction of positive samples, our WCL differs from the approach used in [1].** Based on our discovery that different networks can capture different views, we aim for a more diverse interaction of views during contrastive learning. In contrast to [1], where one of the augmented images are fed into the supernet while another augmented images are directed to networks of different widths, all our networks can observe both two of distinct data augmentations to construct the positive samples. By jointly combining them for contrastive learning, we enhance the diversity of views, fostering knowledge interaction among networks of different widths. Finally, in the results of previous contrastive learning based slimmable networks, the supernet in the slimmable network may be influenced by the smaller networks, resulting in its performance degradation compared to individual training. By contrast, **WCL enables smaller networks to complement the knowledge of larger networks**, ultimately promoting the improvement of the performance of the larger networks.
> > > >
> > > > Secondly, the LCR is different from the loss in slimmable network. Notably, **we are the first to use hard labels to replace the soft labels commonly used** in related methods like [1] for supervision in slimmable network. We generate pseudo-labels by passing weakly augmented images through the largest network, and supervise all width of networks (including supernet itself) on the predictions obtained from strongly augmented images. This is because, in test-time training, only noisy pseudo-labels can be utilized to regularize prediction consistency between supernet and sub-nets, which is in contrast to the previous works that possess ground-truth labels. As a consequence, the entropy minimization effect brought by hard labels becomes more favored than the dark knowledge contained in the soft ones. Through comparison of our SlimTTT using soft labels and hard labels below on CIFAR10-C, we empirically demonstrate the effectiveness of our LCR using hard labels.
> > > >
> > > > | Method  | R-50 (1.0×) | R-50 (0.75×) | R-50 (0.5×) | R-50 (0.25×) |
> > > > | --- | --- | --- | --- | --- |
> > > > | Soft Labels | 8.78 | 9.25 | 9.98 | 11.91 |
> > > > | Hard Labels | **8.33** | **8.63** | **9.12** | **10.68** |
> > > >
> > > > Thirdly, although GFA is a method adopted in previous works, we extended it to multiple width networks. Due to the effectiveness of our consistency constraints, we do not need to perform alignment at the classwise or self-supervised head levels, as done in previous methods. This improves the efficiency of our approach, making it simple yet effective. In addition, ensemble techniques are not universally applicable in all cases, as demonstrated in [3]. Therefore, we do not treat ensemble as a widely used trick. Inspired by [3], we theoretically and visually prove that networks of different widths have the ability to capture multi-view features of images. This finding ensures the effectiveness of using ensemble techniques in our approach.
> > > >
> > > > 3. **The differences of the motivation of our method and unsupervised method as [1].**
> > > >
> > > > Our proposed SlimTTT is not a simple application of Slimmable network on Test-time training. Instead, it is well-motivated to improve the practicability of TTT, with all its modules carefully designed to adapt for the setting.  First, through an analysis of the TTT application scenarios, we identified its limitations when resources are constrained or dynamically changing. Additionally, we theoretically and visually demonstrated that sub-networks of different widths in slimmable networks can capture distinct views. These views naturally adapt during the TTT phase through contrastive learning. With the assistance of our designed techniques, WCL, LCR, and GFA, the model learns more robust features in both online and offline TTT processes. The coexistence of multi-width networks effectively addresses challenges faced by previous TTT methods in resource-constrained and dynamically changing environments, expanding the applicability of TTT.

---

> > > > > ### Author Response · Authors · 2023-11-20
> > > > >
> > > > > **Q2:** Table 2 shows that GFA gives most of the improvements. It is proposed in previous works and not clearly explained how it is applied in the proposed method.
> > > > >
> > > > > **A2:** GFA is a commonly used technique in prior TTT methods, however, they only apply it on a single network. We extended it to multiple width networks by aligning the global test data distribution of different width to the source one of the same width using KL-Divergence. The GFA in our method is applied to ensure that the test features do not deviate far from the source feature distribution during the whole adaptation process and it can apply a restriction over the multi-view features for each width of sub-networks and ensure the correctness of the multi-view representations. The reason we align the test distribution of each width with the corresponding source distribution, rather than imposing consistency constraints, is because we want the networks to retain their unique multi-view characteristics without excessively converging. This ensures the effectiveness of our WCL knowledge interaction and ensemble technique. We conduct the following experiment on CIFAR10-C (Guassian_Noise) to compare SlimTTT using our GFA with the GFA applied on a single network (GFA (single)) and imposing GFA consistency on self-supervised head (GFA (ssh)). The results show that our GFA can outperform the other forms of GFA, which fully leveraging the characteristics of each width of networks.
> > > > >
> > > > > | Method | R-50 (1.0×) | R-50 (0.75×) | R-50 (0.5×) | R-50 (0.25×) |
> > > > > | --- | --- | --- | --- | --- |
> > > > > | GFA (single) | 12.63 | 13.39 | 13.58 | 15.55 |
> > > > > | GFA (ssh) | 11.79 | 12.39 | 13.03 | 15.04 |
> > > > > | GFA (ours) | **10.41** | **11.04** | **11.80** | **13.96** |

---

### Author Response · Authors · 2023-11-19

Dear AC and Reviewers,

We extend our heartfelt appreciation to to all reviewers for their insightful questions, valuable comments, and constructive suggestions. We are encouraged by the recognition of our efficient test-time training setting (noted by 1QhS, aCQv) and clear explanation (highlighted by by2E, aCQv). Our paper provide new insight of the multi-view findings (noted by yJHx) and achieve good performance on several benchmarks (noted by all reviewers).

In response to the thoughtful feedback and suggestions from the reviewers, we have delved deeper into our investigation, making significant enhancements. We have incorporated the following discussions and experiments:

- The comparison with un-superivsed/self-supervised setting (A1 for by2E).
- The comparison with MutualNet about multi-view (A1 for by2E).
- Considering experiments where the available compute changes frequently during test-time training (A1 for 1QhS).
- Explaining the motivation for SlimTTT (A1 for yJHx).
- Applying SlimTTT to online test-time training setting (A3 for yJHx).
- Explaining the “Efficient” and basic assumptions of our SlimTTT (A1 and A2 for aCQv).

Enthusiastically inspired by the insightful remarks and suggestions from the reviewers, we have uncovered further insights into the ability of online dynamic adaptation during test-time training phase of our SlimTTT approach. We hope to have been able to address all of the reviewers' questions and concerns and remain available to provide answers for any follow-up inquiries.

Warm regards,

Authors

---

### Meta-Review · Area_Chair_tHEa · 2023-12-05

**Metareview:**

This paper proposes a method for test-time training (TTT) which is the offline fine-training of a source model using unlabeled data from a new domain, followed by inference on actual test inputs from the new domain.

The proposed method is an extension of the slimmable networks idea adapted for the TTT setting which is unsupervised. Albeit not in the TTT setting, unsupervised extensions of slimmable networks have been proposed in prior work as well, as pointed out by one of the reviewers.

The reviewers also expressed concerns about the idea being basically a combination of unsupervised source-free domain adaptation and slimmable networks. There are also concerns regarding efficiency of SlimTTT as compared to methods like TENT. The reviewers were not entirely convinced why slimmable networks would be appropriate.

The authors' response was considered and discussed but the reviewers' concerns remained. Therefore, in the end, it was felt that the paper in its current form is not ready to be published. The authors are advised to address the concerns raised by the reviewers and incorporate the suggested experiments, and resubmit the work to another venue.

**Justification For Why Not Higher Score:**

Several issues pointed out by the reviewers regarding the novelty and the motivation behind using slimmable networks

**Justification For Why Not Lower Score:**

N/A

---

### Decision · Program_Chairs · 2024-01-16

Reject